# Multi-Level Local SGD: Distributed SGD for Heterogeneous Hierarchical Networks

**Timothy Castiglia, Anirban Das, and Stacy Patterson**[*]

## Abstract

We propose Multi-Level Local SGD, a distributed stochastic gradient method for learning a smooth, non-convex objective in a multi-level communication network with heterogeneous workers. Our network model consists of a set of disjoint sub-networks, with a single hub and multiple workers; further, workers may have different operating rates. The hubs exchange information with one another via a connected, but not necessarily complete, communication network. In our algorithm, sub-networks execute a distributed SGD algorithm, using a hub-and-spoke paradigm, and the hubs periodically average their models with neighboring hubs. We first provide a unified mathematical framework that describes the Multi-Level Local SGD algorithm. We then present a theoretical analysis of the algorithm; our analysis shows the dependence of the convergence error on the worker node heterogeneity, hub network topology, and the number of local, sub-network, and global iterations. We illustrate the effectiveness of our algorithm in a multi-level network with slow workers via simulation-based experiments.

## 1 Introduction

Stochastic Gradient Descent (SGD) is a key algorithm in modern Machine Learning and optimization (Amari, 1993). To support distributed data as well as reduce training time, Zinkevich et al. (2010) introduced a distributed form of SGD. Traditionally, distributed SGD is run within a hub-and-spoke network model: a central parameter server (hub) coordinates with worker nodes. At each iteration, the hub sends a model to the workers. The workers each train on their local data, taking a gradient step, then return their locally trained model to the hub to be averaged. Distributed SGD can be an efficient training mechanism when message latency is low between the hub and workers, allowing gradient updates to be transmitted quickly at each iteration. However, as noted in Moritz et al. (2016), message transmission latency is often high in distributed settings, which causes a large increase in overall training time. A practical way to reduce this communication overhead is to allow the workers to take multiple local gradient steps before communicating their local models to the hub. This form of distributed SGD is referred to as Local SGD (Lin et al., 2018; Stich, 2019). There is a large body of work that analyzes the convergence of Local SGD and the benefits of multiple local training rounds (McMahan et al., 2017; Wang & Joshi, 2018; Li et al., 2019).

Local SGD is not applicable to all scenarios. Workers may be heterogeneous in terms of their computing capabilities, and thus the time required for local training is not uniform. For this reason, it can be either costly or impossible for workers to train in a fully synchronous manner, as stragglers may hold up global computation. However, the vast majority of previous work uses a synchronous model, where all clients train for the same number of rounds before sending updates to the hub (Dean et al., 2012; Ho et al., 2013; Cipar et al., 2013). Further, most works assume a hub-and-spoke model, but this does not capture many real world settings. For example, devices in an ad-hoc network may not all be able to communicate to a central hub in a single hop due to network or communication range limitations. In such settings, a multi-level communication network model may be beneficial. In flying ad-hoc networks (FANETs), a network architecture has been proposed to improve scalability by partitioning the UAVs into mission areas (Bekmezci et al., 2013). Here, clusters of UAVs have their own clusterheads, or hubs, and these hubs communicate through an upper level network, e.g., via satellite. Multi-level networks have also been utilized in Fog and Edge computing, a paradigm de-

---

[*]T. Castiglia, A. Das, and S. Patterson are with the Department of Computer Science, Rensselaer Polytechnic Institute, 110 8th St, Troy, NY 12180, `castit@rpi.edu`, `dasa2@rpi.edu`, `sep@cs.rpi.edu`.

signed to improve data aggregation and analysis in wireless sensor networks, autonomous vehicles, power systems, and more (Bonomi et al., 2012; Laboratory, 2017; Satyanarayanan, 2017).

Motivated by these observations, we propose Multi-Level Local SGD (MLL-SGD), a distributed learning algorithm for heterogeneous multi-level networks. Specifically, we consider a two-level network structure. The lower level consists of a disjoint set of hub-and-spoke *sub-networks*, each with a single hub server and a set of workers. The upper level network consists of a connected, but not necessarily complete, *hub network* by which the hubs communicate. For example, in a Fog Computing application, the sub-network workers may be edge devices connected to their local data center, and the data centers act as hubs communicating over a decentralized network. Each sub-network runs one or more Local SGD rounds, in which its workers train for a local training period, followed by model averaging at the sub-network's hub. Periodically, the hubs average their models with neighbors in the hub network. We model heterogeneous workers using a stochastic approach; each worker executes a local training iteration in each time step with a probability proportional to its computational resources. Thus, different workers may take different numbers of gradient steps within each local training period. Note since MLL-SGD averages every local training period, regardless of how many gradient steps each worker takes, slow workers do not slow algorithm execution.

We prove the convergence of MLL-SGD for smooth and potentially non-convex loss functions. We assume data is distributed in an IID manner to all workers. Further, we analyze the relationship between the convergence error and algorithm parameters and find that, for a fixed step size, the error is quadratic in the number of local training iterations and the number of sub-network training iterations, and linear in the average worker operating rate. Our algorithm and analysis are general enough to encompass several variations of SGD as special cases, including classical SGD (Amari, 1993), SGD with weighted workers (McMahan et al., 2017), and Decentralized Local SGD with an arbitrary hub communication network (Wang & Joshi, 2018). Our work provides novel analysis of a distributed learning algorithm in a multi-level network model with heterogeneous workers.

The specific contributions of this paper are as follows. 1) We formalize the multi-level network model with heterogeneous workers, and we define the MLL-SGD algorithm for training models in such a network. 2) We provide theoretical analysis of the convergence guarantees of MLL-SGD with heterogeneous workers. 3) We present an experimental evaluation that highlights our theoretical convergence guarantees. The experiments show that in multi-level networks, MLL-SGD achieves a marked improvement in convergence rate over algorithms that do not exploit the network hierarchy. Further, when workers have heterogeneous operating rates, MLL-SGD converges more quickly than algorithms that require all workers to execute the same number of training steps in each local training period.

The rest of the paper is structured as follows. In Section 2, we discuss related work. Section 3 introduces the system model and problem formulation. We describe MLL-SGD in Section 4, and we present our main theoretical results in Section 5. Proofs of these results are deferred to the appendix. We provide experimental results in Section 6. Finally, we conclude in Section 7.

## 2 RELATED WORK

Distributed SGD is a well studied subject in Machine Learning. Zinkevich et al. (2010) introduced parallel SGD in a hub-and-spoke model. Variations on Local SGD in the hub-and-spoke model have been studied in several works (Moritz et al., 2016; Zhang et al., 2016; McMahan et al., 2017). Many works have provided convergence bounds of SGD within this model (Wang et al., 2019b; Li et al., 2019). There is also a large body of work on decentralized approaches for optimization using gradient based methods, dual averaging, and deep learning (Tsitsiklis et al., 1986; Jin et al., 2016; Wang et al., 2019a). These previous works, however, do not address a multi-level network structure.

In practice, workers may be heterogeneous in nature, which means that they may execute training iterations at different rates. Lian et al. (2017) addressed this heterogeneity by defining a gossip-based asynchronous SGD algorithm. In Stich (2019), workers are modeled to take gradient steps at an arbitrary subset of all iterations. However, neither of these works address a multi-level network model. Grouping-SGD (Jiang et al., 2019) considers a scenario where workers can be clustered into groups, for example, based on their operating rates. Workers within a group train in a synchronous manner, while the training across different groups may be asynchronous. The system model differs

significantly from that in MLL-SGD in that as the model parameters are partitioned vertically across multiple hubs, and workers communicate with every hub.

Several recent works analyze Hierarchical Local SGD (HL-SGD), an algorithm for training a model in a hierarchical network. Different from MLL-SGD, HL-SGD assumes the hub network topology is a hub-and-spoke and also that workers are homogeneous. Zhou & Cong (2019) and Liu et al. (2020) analyze the convergence error of HL-SGD, while Abad et al. (2020) analyzes convergence time. Unlike HL-SGD, MLL-SGD accounts for an arbitrary hub communication graph, and MLL-SGD algorithm execution does not slow down in the presence of heterogeneous worker operating rates.

Several other works seek to encapsulate many variations of SGD under a single framework. Koloskova et al. (2020) created a generalized model that considers a gossip-based decentralized SGD algorithm where the communication network is time-varying. However, this work does not account for a multi-level network model nor worker heterogeneity. Wang et al. introduced the Co-operative SGD framework (Wang & Joshi, 2018), a model that includes communication reduction through local SGD steps and decentralized mixing between homogeneous workers. Cooperative SGD also allows for auxiliary variables. These auxiliary variables can be used to model SGD in a multi-level network, but only when sub-network averaging is immediately followed by hubs averaging with their neighbors in the hub network. Our model is more general; it considers heterogeneous workers and it allows for an arbitrary number of averaging rounds within each sub-network between averaging rounds across sub-networks, which is more practical in multi-level networks where inter-hub communication is slow or costly.

## 3  System Model and Problem Formulation

In this section, we introduce our system model, the objective function that we seek to minimize, and the assumptions we make about the function.

We consider a set of $D$ sub-networks $\mathcal{D} = \{1, \ldots, D\}$. Each sub-network $d \in \mathcal{D}$ has a single hub and a set of workers $\mathcal{M}^{(d)}$, with $|\mathcal{M}^{(d)}| = N^{(d)}$. Workers in $\mathcal{M}^{(d)}$ only communicate with their own hub and not with any other workers or hubs. We define the set of all workers in the system as $\mathcal{M} = \bigcup_{d=1}^{D} \mathcal{M}^{(d)}$. Let $|\mathcal{M}| = N$. Each worker $i$ holds a set $\mathcal{S}^{(i)}$ of local training data. Let $\mathcal{S} = \bigcup_{i=1}^{N} \mathcal{S}^{(i)}$. The set of all $D$ hubs is denoted $\mathcal{C}$. The hubs communicate with one another via an undirected, connected communication graph $G = (\mathcal{C}, E)$. Let $\mathcal{N}_d = \{j \mid e_{d,j} \in E\}$ denote the set of neighbors of the hub in sub-network $d$ in the hub graph $G$.

Let the model parameters be denoted by $\boldsymbol{x} \in \mathbb{R}^n$. Our goal is to find an $\boldsymbol{x}$ that minimizes the following objective function over the training set:

$$F(\boldsymbol{x}) = \frac{1}{|\mathcal{S}|} \sum_{s \in \mathcal{S}} f(\boldsymbol{x}; s) \tag{1}$$

where $f(\cdot)$ is the loss function. The workers collaboratively minimize this loss function, in part by executing local iterations of SGD over their training sets. For each executed local iteration, a worker samples a mini-batch of data uniformly at random from its local data. Let $\xi$ be a randomly sampled mini-batch of data and let $g(\boldsymbol{x}; \xi) = \frac{1}{|\xi|} \sum_{s \in \xi} \nabla f(\boldsymbol{x}; s)$ be the mini-batch gradient. For simplicity, we use $g(\boldsymbol{x})$ instead of $g(\boldsymbol{x}; \xi)$ from here on.

**Assumption 1.** *The objective function and the mini-batch gradients satisfy the following:*

*1a  The objective function $F : \mathbb{R}^n \to \mathbb{R}$ is continuously differentiable, and the gradient is Lipschitz with constant $L > 0$, i.e., $\|\nabla F(\boldsymbol{x}) - \nabla F(\boldsymbol{y})\|_2 \leq L\|\boldsymbol{x} - \boldsymbol{y}\|_2$ for all $\boldsymbol{x}, \boldsymbol{y} \in \mathbb{R}^n$.*

*1b  The function $F$ is lower bounded, i.e., $F(\boldsymbol{x}) \geq F_{inf} > -\infty$ for all $\boldsymbol{x} \in \mathbb{R}^n$.*

*1c  The mini-batch gradients are unbiased, i.e., $\mathbb{E}_{\xi|\boldsymbol{x}}[g(\boldsymbol{x})] = \nabla F(\boldsymbol{x})$ for all $\boldsymbol{x} \in \mathbb{R}^n$.*

*1d  There exist scalars $\beta \geq 0$ and $\sigma \geq 0$ such that $\mathbb{E}_{\xi|\boldsymbol{x}}\|g(\boldsymbol{x}) - \nabla F(\boldsymbol{x})\|_2^2 \leq \beta\|\nabla F(\boldsymbol{x})\|_2^2 + \sigma^2$ for all $\boldsymbol{x} \in \mathbb{R}^n$.*

Assumption 1a requires that the gradients do not change too rapidly, and Assumption 1b requires that our objective function is lower bounded by some $F_{inf}$. Assumptions 1c and 1d assume that

---

**Algorithm 1** Multi-Level Local SGD

---

1: **Initialize:** $\boldsymbol{y}_1^{(d)}$ for hubs $d = 1, \ldots, D$
2: **for** $k = 1, \ldots, K$ **do**
3:      **parallel for** $d \in \mathcal{D}$ **do**
4:          **parallel for** $i \in \mathcal{M}^{(d)}$ **do**
5:              $\boldsymbol{x}_k^{(i)} \leftarrow \boldsymbol{y}_k^{(d)}$                     $\triangleright$ Workers receive updated model from hub
6:              **for** $j = k, \ldots, k + \tau - 1$ **do**
7:                  $\boldsymbol{x}_{k+1}^{(i)} \leftarrow \boldsymbol{x}_k^{(i)} - \eta \boldsymbol{g}_k^{(i)}$              $\triangleright$ Local iteration (probabilistic)
8:              **end for**
9:          **end parallel for**
10:          $\boldsymbol{z}^{(d)} \leftarrow \sum_{i \in \mathcal{M}^{(d)}} v^{(i)} \boldsymbol{x}_{k+1}^{(i)}$        $\triangleright$ Hub $d$ computes average of its workers' models
11:          **if** $k \bmod q \cdot \tau = 0$ **then**
12:              $\boldsymbol{y}_{k+1}^{(d)} \leftarrow \sum_{j \in \mathcal{N}^{(d)}} \mathbf{H}_{j,d} \boldsymbol{z}^{(j)}$      $\triangleright$ Hub $d$ averages its model with neighboring hubs
13:          **else**
14:              $\boldsymbol{y}_{k+1}^{(d)} \leftarrow \boldsymbol{z}^{(d)}$
15:          **end if**
16:      **end parallel for**
17: **end for**

---

the local data at each worker can be used as an unbiased estimate for the full dataset with the same bounded variance. These assumptions are common in convergence analysis of SGD algorithms (e.g., Bottou et al. (2018)).

## 4 ALGORITHM

We now present our Multi-Level Local SGD (MLL-SGD) algorithm. The pseudocode is shown in Algorithm 1. Each sub-network trains in parallel and, periodically, the hubs average their models with neighboring hubs. The steps corresponding to Local SGD are shown in lines 5-10. Each hub and worker stores a copy of the model. For worker $i \in \mathcal{M}^{(d)}$, we denote its copy of the local model by $\boldsymbol{x}^{(i)}$. We denote the model at hub $d$ by $\boldsymbol{y}^{(d)}$. The hub first sends its model to its workers, and the workers update their local models to match their hub's model. Workers then execute multiple local training iterations, shown in line 7, to refine their local models independently. To represent the different rates of computation at each worker, we use a probabilistic approach. We assume that, in expectation, a worker $i$ execute $\tau^{(i)}$ local iterations for every $\tau$ time steps ($\tau^{(i)} \leq \tau$). We thus define the N-vector $\boldsymbol{p}$ where each entry $\boldsymbol{p}_i = \frac{\tau^{(i)}}{\tau}$ is the probability with which worker $i$ executes a local gradient step in each iteration $k$. Worker $i$ updates its local model at iteration $k$ as follows:

$$\boldsymbol{x}_{k+1}^{(i)} = \boldsymbol{x}_k^{(i)} - \eta \boldsymbol{g}_k^{(i)} \tag{2}$$

where $\eta$ is the step size and $\boldsymbol{g}_k^{(i)}$ is a random variable such that

$$\boldsymbol{g}_k^{(i)} = \begin{cases} g(\boldsymbol{x}_k^{(i)}) & \text{w/ probability } \boldsymbol{p}_i \\ \mathbf{0} & \text{w/ probability } 1 - \boldsymbol{p}_i. \end{cases} \tag{3}$$

After $\tau$ time steps, the hub updates its model based on the models of its workers (line 10). For each worker $i$, we assign a positive weight $w^{(i)}$. Let $v^{(i)}$ be the weight for worker $i$ normalized within its sub-network: $v^{(i)} = \frac{w^{(i)}}{\sum_{j \in \mathcal{M}^{(d(i))}} w^{(j)}}$, where $d(i)$ denotes the sub-network of worker $i$. Each hub's updates its model to be a weighted average over the workers' models in its sub-network: $\boldsymbol{y}^{(d)} = \sum_{i \in \mathcal{M}^{(d)}} v^{(i)} \boldsymbol{x}^{(i)}$. Weights may be assigned for different reasons. If all worker gradients are treated equally, then $w^{(i)} = 1$ and $v^{(i)} = \frac{1}{N^{(d(i))}}$. We may also weight a worker's gradient proportional to its local dataset size, in which case $w^{(i)} = |\mathcal{S}^{(i)}|$ and $v^{(i)} = \frac{|\mathcal{S}^{(i)}|}{\sum_{r \in \mathcal{M}^{(d(i))}} |\mathcal{S}^{(r)}|}$. The latter approach is used in Federated Averaging (McMahan et al., 2017).

After $q$ iterations of Local SGD in each sub-network ($q \cdot \tau$ time steps), the hubs average their models with their neighbors in the hub communication network (line 12). The weight assigned to each hub's

model is defined by a $D \times D$ matrix $\mathbf{H}$ so that:

$$\boldsymbol{y}^{(d)} = \sum_{j \in \mathcal{N}^{(d)}} \mathbf{H}_{j,d} \boldsymbol{y}^{(j)}. \tag{4}$$

Define the total weight in the network to be $w_{tot} = \sum_{i \in \mathcal{M}} w^{(i)}$. Let $\boldsymbol{b}$ be a $D$-vector with each component $d$ given by $\boldsymbol{b}_d = (\sum_{i \in \mathcal{M}^{(d)}} w^{(i)})/w_{tot}$. We assume $\mathbf{H}$ meets the following requirements.

**Assumption 2.** *The matrix $\mathbf{H}$ satisfies the following:*

   *2a If $(i,j) \in E$, then $\mathbf{H}_{i,j} > 0$. Otherwise, $\mathbf{H}_{i,j} = 0$.*

   *2b $\mathbf{H}$ is column stochastic, i.e., $\sum_{i=1}^{D} \mathbf{H}_{i,j} = 1$.*

   *2c For all $i, j \in \mathcal{D}$, we have $\boldsymbol{b}_i \mathbf{H}_{i,j} = \boldsymbol{b}_j \mathbf{H}_{j,i}$.*

Assumption 2 implies that $\mathbf{H}$ has one as a simple eigenvalue, with corresponding right eigenvector $\boldsymbol{b}$ and left eigenvector $\mathbf{1}_D$. Further, all of its other eigenvalues have magnitude strictly less than 1 (since $G$ is connected) (Rotaru & Nägeli, 2004). By defining $\mathbf{H}$ in this way, we ensure that the contributions from the workers' gradients in each hub are incorporated in proportion to the workers' weights. This weighted averaging approach allows us to naturally extend Federated Averaging to the multi-level network model.

## 5 ANALYSIS

We note that hubs are essentially stateless in MLL-SGD, as the hub models are copied to all workers after each sub-network or hub averaging. Thus, our analysis focuses on how worker models evolve. We first present an equivalent formulation of the MLL-SGD algorithm in terms of the evolution of the worker models. We then present our main result on the convergence of MLL-SGD.

The system behavior can be summarized by the following update rule for worker models:

$$\mathbf{X}_{k+1} = (\mathbf{X}_k - \eta \, \mathbf{G}_k) \, \mathbf{T}_k \tag{5}$$

where $n \times N$ matrix $\mathbf{X}_k = [\boldsymbol{x}_k^{(1)}, \dots, \boldsymbol{x}_k^{(N)}]$, $n \times N$ matrix $\mathbf{G}_k = [\boldsymbol{g}_k^{(1)}, \dots, \boldsymbol{g}_k^{(N)}]$, and $N \times N$ matrix $\mathbf{T}_k$ is a time-varying operator that captures the three stages in MLL-SGD: local iterations, hub-and-spoke averaging within each sub-network, and averaging across the hub network. We define $\mathbf{T}_k$ as follows:

$$\mathbf{T}_k = \begin{cases} \mathbf{Z} & \text{if } k \bmod q\tau = 0 \\ \mathbf{V} & \text{if } k \bmod \tau = 0 \text{ and } k \bmod q\tau \neq 0 \\ \mathbf{I} & \text{otherwise.} \end{cases} \tag{6}$$

For local iterations, $\mathbf{T}_k = \mathbf{I}$, as there are no interactions between workers or hubs. For sub-network averaging, $\mathbf{V}$ is an $N \times N$ block diagonal matrix, with each block $\mathbf{V}^{(d)}$ corresponding to a single sub-network $d$. The matrix $\mathbf{V}^{(d)}$ is an $N^{(d)} \times N^{(d)}$ matrix where each entry is $\mathbf{V}_{i,j}^{(d)} = v^{(i)}$. Finally, we define an $N \times N$ matrix $\mathbf{Z}$ that captures the sub-network averaging and hub network averaging in one operation that involves all workers. The components of $\mathbf{Z}$ are given by

$$\mathbf{Z}_{i,j} = \mathbf{H}_{d(i),d(j)} v^{(i)}. \tag{7}$$

Let $\boldsymbol{a}$ be an $N$-vector with each component $\boldsymbol{a}_i = \frac{w^{(i)}}{w_{tot}}$ representing the weight of worker $i$, normalized over all worker weights. We observe that $\mathbf{Z}$ and $\mathbf{V}$ satisfy the following: each have a right eigenvector of $\boldsymbol{a}$ and left eigenvector of $\mathbf{1}_N^T$ with eigenvalue 1 and all other eigenvalues have magnitude strictly less than 1. The proof of these properties can be found in the appendix. These properties are necessary (but not sufficient) to ensure that the worker models converge to a consensus model, where each worker's updates have been incorporated according to the worker's weight.

As is common, we study an averaged model over all workers in the system (Yuan et al., 2016; Wang & Joshi, 2018). Specifically, we define a weighted average model:

$$\boldsymbol{u}_k = \mathbf{X}_k \, \boldsymbol{a}. \tag{8}$$

We identify the recurrence relation of $\boldsymbol{u}_k$. If we multiply $\boldsymbol{a}$ on both sides of (5):

$$\mathbf{X}_{k+1}\,\boldsymbol{a} = (\mathbf{X}_k - \eta\,\mathbf{G}_k)\,\mathbf{T}_k\,\boldsymbol{a} \tag{9}$$

$$\boldsymbol{u}_{k+1} = \boldsymbol{u}_k - \eta\,\mathbf{G}_k\,\boldsymbol{a} \tag{10}$$

$$\boldsymbol{u}_{k+1} = \boldsymbol{u}_k - \eta\sum_{i=1}^{N}\boldsymbol{a}_i\boldsymbol{g}_k^{(i)} \tag{11}$$

where (10) follows from $\boldsymbol{a}$ being a right eigenvector of $\mathbf{V}$ and $\mathbf{Z}$ with eigenvalue 1. We note that $\boldsymbol{u}_k$ is updated via a stochastic gradient descent step using a weighted average of several mini-batch gradients. Since $F(\cdot)$ may be non-convex, SGD may converge to a local minimum or saddle point. Thus, we study the gradients of $\boldsymbol{u}_k$ as $k$ increases.

We next provide the main theoretical result of the paper.

**Theorem 1.** *Under Assumptions 1 and 2, if $\eta$ satisfies the following for all $i \in \mathcal{M}$:*

$$(4\boldsymbol{p}_i - \boldsymbol{p}_i^2 - 2) \geq \eta L\left(\boldsymbol{a}_i\boldsymbol{p}_i(\beta + 1) - \boldsymbol{a}_i\boldsymbol{p}_i^2 + \boldsymbol{p}_i^2\right) + 8L^2\eta^2q^2\tau^2\Gamma \tag{12}$$

*where $\Gamma = \frac{\zeta}{1-\zeta^2} + \frac{2}{1-\zeta} + \frac{\zeta}{(1-\zeta)^2}$ and $\zeta = \max\{|\lambda_2(\mathbf{H})|, |\lambda_N(\mathbf{H})|\}$, then the expected square norm of the average model gradient, averaged over $K$ iterations, is bounded as follows:*

$$\mathbb{E}\left[\frac{1}{K}\sum_{k=1}^{K}\|\nabla F(\boldsymbol{u}_k)\|_2^2\right] \leq \frac{2\left(F(\boldsymbol{x}_1) - F_{inf}\right)}{\eta K} + \sigma^2\eta L\sum_{i=1}^{N}\boldsymbol{a}_i^2\boldsymbol{p}_i$$

$$+ 4L^2\eta^2\sigma^2q^3\tau^3\left(\frac{1}{q\tau} - \frac{1}{K}\right)\left(\frac{\zeta^2}{1-\zeta^2} + \frac{2\zeta}{1-\zeta} + \frac{1}{(1-\zeta)^2}\right)\mathrm{P}$$

$$+ 4L^2\eta^2\sigma^2\left(\frac{2-\zeta}{1-\zeta}\right)\left(\tau^2\frac{(q-1)(2q+1)}{6} + \frac{(\tau-1)(2\tau+1)}{6}\right)\mathrm{P} \tag{13}$$

$$\xrightarrow{K\to\infty}\quad \sigma^2\eta L\sum_{i=1}^{N}\boldsymbol{a}_i^2\boldsymbol{p}_i + 4L^2\eta^2\sigma^2q^2\tau^2\left(\frac{\zeta^2}{1-\zeta^2} + \frac{2\zeta}{1-\zeta} + \frac{1}{(1-\zeta)^2}\right)\mathrm{P}$$

$$+ 4L^2\eta^2\sigma^2\left(\frac{2-\zeta}{1-\zeta}\right)\left(\tau^2\frac{(q-1)(2q+1)}{6} + \frac{(\tau-1)(2\tau+1)}{6}\right)\mathrm{P} \tag{14}$$

*where $\mathrm{P} = \sum_{i=1}^{N}\boldsymbol{a}_i\boldsymbol{p}_i$.*

The proof of Theorem 1 is provided in the appendix. The first term in (13) is the same as in centralized SGD (Bottou et al., 2018). As $K \to \infty$, this term goes to zero. The second term is similar to centralized SGD as well. If the stochastic gradients have high variance, then the convergence error will be larger. This term is also related to the convergence error in distributed SGD (Bottou et al., 2018), which is equivalent to MLL-SGD when there is one sub-network, $q = \tau = 1$, $\boldsymbol{a}_i = 1/N$, and $\boldsymbol{p}_i = 1$ for all $i$. MLL-SGD has a dependence on the probabilities of gradient steps and worker weights, replacing the $\frac{1}{N}$ in the equivalent term in distributed SGD.

The third and fourth terms in (13) are additive errors that depend on the topology of the hub network. The value of $\zeta$ is given by the second largest eigenvalue of $\mathbf{H}$, by magnitude, which is an indication of the sparsity of the hub network. When worker weights are uniform, a fully connected hub graph $G$ will have $\zeta = 0$, while a sparse $G$ will typically have $\zeta$ close to 1. It is interesting to note that $\zeta$ only depends on $\mathbf{H}$, and not $\mathbf{Z}$ or $\mathbf{V}$, meaning the convergence error does not depend on how worker weights are distributed within sub-networks.

We also note the third and fourth terms depend on $\mathrm{P}$, the weighted average probability of the workers. The convergence error increases as the average worker operating rate increases. This relation is expected as more local iterations will increase convergence error (Wang & Joshi, 2018). It is interesting to note that the convergence error does not depend on the distribution of $\boldsymbol{p}$, meaning that a skewed and uniform distribution with the same average probability would have the same convergence error. We observe that the condition on $\eta$ in (12) cannot always be satisfied given certain probabilities. Specifically, when there exists a $\boldsymbol{p}_i \leq 2 - \sqrt{2} \approx 0.59$, then the left-hand side will be non-positive, and the inequality can no longer be satisfied. Although this may be a conservative bound, intuitively, when $\boldsymbol{p}_i$'s are below this threshold, the algorithm may not make sufficient progress in each time step to guarantee convergence.

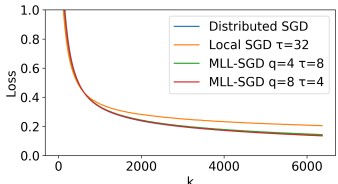

(a) Training loss of CNN trained on EMNIST.

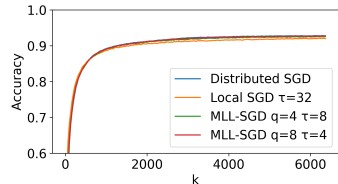

(b) Test accuracy of CNN trained on EMNIST.

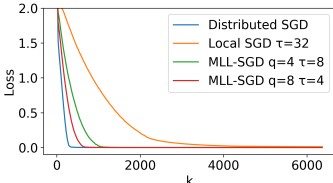

(c) Training loss of ResNet-18 trained on CIFAR-10.

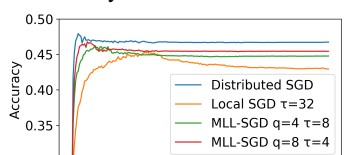

(d) Test accuracy of ResNet-18 trained on CIFAR-10.

Figure 1: Effect of a hierarchy with different values of $\tau$ and $q$.

The third and fourth terms also grow with $q$ and $\tau$, the number of local iterations per hub network averaging and sub-network averaging steps, respectively. The longer workers train locally without reconciling their models, the more their models will diverge, leading to larger convergence error. We can see that $\tau$ plays a slightly larger role in convergence error than $q$. For a given $q \cdot \tau$, meaning a given number of time steps between hub averaging steps, a larger $\tau$ leads to higher convergence error than a larger $q$ would. Thus, there is a slight penalty to performing more local iterations between sub-network averaging steps. We explore this more in Section 6.

We note that when setting $a_i = 1/N$ and $p_i = 1$ for all workers $i$, and setting $q = 1$, MLL-SGD reduces to Cooperative SGD. However, the bound in Theorem 1 differs from that of Cooperative SGD. Specifically, Theorem 1 has error terms dependent on $\tau^2$ as opposed to $\tau$ in Cooperative SGD. This discrepancy is due to accommodating all possible values of $p_i$. More details can be found in Appendix C.4.

In the following corollary, we analyze the convergence rate of Algorithm 1 when $\eta = \frac{1}{L\sqrt{K}}$.

**Corollary 1.** *Let $\eta = \frac{1}{L\sqrt{K}}$ and let $q^2\tau^2 \leq \sqrt{K}$. If $q\tau < K$, then*

$$\mathbb{E}\left[\frac{1}{K}\sum_{k=1}^{K}\|\nabla F(\boldsymbol{u}_k)\|_2^2\right] \leq O\left(\frac{L}{\sqrt{K}}\right)(F(\boldsymbol{x}_1) - F_{inf}]) + O\left(\frac{\sigma^2}{\sqrt{K}}\right) \tag{15}$$

Under the conditions given in Corollary 1, MLL-SGD achieves the same asymptotic convergence rate as Local SGD and HL-SGD.

## 6 EXPERIMENTS

In this section, we show the performance of MLL-SGD compared to algorithms that do not account for hierarchy and heterogeneous worker rates. We also explore the impact of the different algorithm parameters that show up in Theorem 1.

We use the EMNIST (Cohen et al., 2017) and CIFAR-10 (Krizhevsky et al., 2009) datasets. For all experiments, we provide results for training a simple Convolutional Neural Network (CNN) on EMNIST and training ResNet-18 on CIFAR-10. The CNN has two convolutional layers and two fully connected layers. We train the CNN with a step size of $0.01$. For ResNet, we use a standard approach of changing the step size from $0.1$ to $0.01$ to $0.001$ over the course of training (He et al., 2016). We conduct experiments using Pytorch 1.4.0 and Python 3.

We compare MLL-SGD with Distributed SGD, Local SGD, and HL-SGD. Distributed SGD is equivalent to MLL-SGD when there is one hub, $q = \tau = 1$, and $a_i = 1/N$ and $p_i = 1$ for all $i$, which means Distributed SGD averages all worker models at every iteration. Thus, we use Distributed

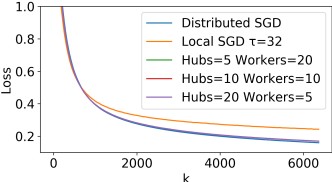

Figure 2: Effect of worker distribution on CNN trained on EMNIST.

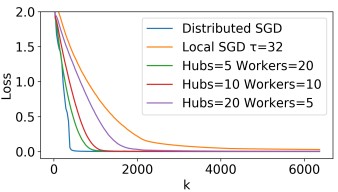

Figure 3: Effect of worker distribution on ResNet-18 trained on CIFAR-10.

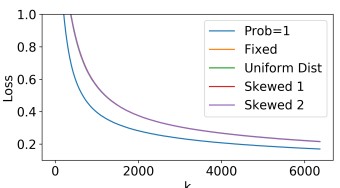

Figure 4: Effect of heterogeneous operating rates on CNN trained on EMNIST.

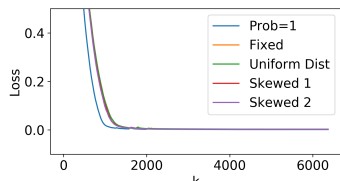

Figure 5: Effect of heterogeneous operating rates on ResNet-18 trained on CIFAR-10.

SGD as a baseline for convergence error and accuracy in some experiments. Local SGD is equivalent to MLL-SGD when $\boldsymbol{a}_i = 1/N$ and $\boldsymbol{p}_i = 1$ for all $i$, when the hub network is fully connected, and $q = 1$. HL-SGD extends Local SGD to allow $q > 1$. For all experiments, we let $\tau = 32$ for Local SGD. We let $q\tau = 32$ for all HL-SGD and MLL-SGD variations to be comparable with Local SGD. In all experiments, we measure training loss and test accuracy of the averaged model $\boldsymbol{u}_k$ every 32 iterations.

We first explore the effect of different values of $\tau$ and $q$ in MLL-SGD. We configure a multi-level network with a fully connected hub network and with 10 hubs, each with 10 workers. We use two configurations for MLL-SGD, one with $\tau = 8$ and $q = 4$, and one with $\tau = 4$ and $q = 8$. Distributed SGD and Local SGD treat the hubs as pass-throughs, and average all workers every iteration and every $\tau$ iterations respectively. Workers are split into five groups of 20 workers each. Each group is assigned a percentage of the full dataset: 5%, 10%, 20%, 25%, and 40%. Workers within a group partition the data evenly. The workers weights are assigned based on dataset sizes. In Figures 1a and 1c we plot the training loss, and in Figures 1b and 1d we plot the test accuracy for the CNN and ResNet, respectively. We observe that as $q$ increases, while keeping $q\tau = 32$, MLL-SGD improves and approaches the Distributed SGD baseline. Thus, increasing the number of sub-network training rounds improves the convergence behavior of MLL-SGD. The benefit is more pronounced in training ResNet on CIFAR.

We next investigate how the number and sizes of the sub-networks impacts the convergence of MLL-SGD. From a pool of 100 workers, we distribute them across 5, 10, and 20 sub-networks. The hub network is a path graph, which yields the largest $\zeta$ while keeping the network connected. This hub network topology the worst-case scenario in terms of the convergence bound. Note that as the number of hubs increases, the larger $\zeta$ becomes. We let $\boldsymbol{a}_i = 1/N$ and $\boldsymbol{p}_i = 1$ for all workers $i$. We set $q = 4$ and $\tau = 8$. We also include results using Local SGD with 1 hub and 100 workers. The results of this experiment are shown in Figures 2 and 3. In the case of the CNN, the difference in training loss is minimal among the MLL-SGD variations. In the case of ResNet we can see that as the number of hubs increase, the convergence rate decreases. This is in line with Theorem 1 since an increased number hubs corresponds with an increased $\zeta$. Interestingly, despite the low hub network connectivity, MLL-SGD outperforms Local SGD. This shows that MLL-SGD still benefits from a hierarchy even when hub connectivity is sparse.

Next, we explore the impact of different distributions of worker operating rates. According to Theorem 1, the average probability across workers plays a role in the error bound. To see if this holds in practice, we compare four different MLL-SGD setups, all of which includes a complete hub network, 10 hubs, each with 10 workers, $\boldsymbol{a}_i = 1/N$, and an average probability amongst workers of 0.55: (i) all workers with a $\boldsymbol{p}_i = 0.55$ (Fixed); (ii) workers in each sub-network with probability ranging from 0.1 to 1 at steps of 0.1 (Uniform Distribution); (iii) 90 workers with $\boldsymbol{p}_i = 0.5$ and 10 workers with $\boldsymbol{p}_i = 1$ (Skewed 1); (iv) 90 workers with $\boldsymbol{p}_i = 0.6$ and 10 workers with $\boldsymbol{p}_i = 0.1$ (Skewed 2). We include a case where all workers have $\boldsymbol{p}_i = 1$ as a baseline (Prob=1). In Figures 4

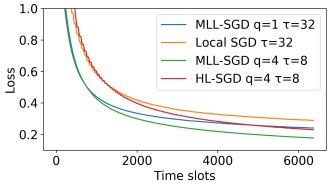

(a) Training loss of CNN with respect to time slots.

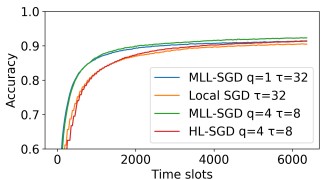

(b) Test accuracy of CNN with respect to time slots.

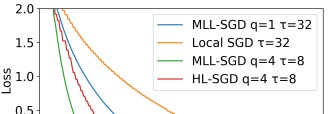

(c) Training loss of ResNet with respect to time slots.

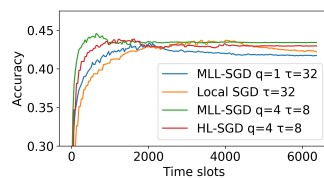

(d) Test accuracy of ResNet with respect to time slots.

Figure 6: Comparing convergence time of Local SGD, HL-SGD, and MLL-SGD.

and 5 we can see that in all cases except the baseline, the convergence rate is similar in both models. This is in line with our theoretical results, since all cases have the same average worker probability.

Finally, we compare the convergence time of MLL-SGD against algorithms that wait for slower workers: Local SGD and HL-SGD. We simulate real-time with time slots. In every time slot, each worker will take a gradient step with a probability $p_i$. Note when $p_i = 1$ for a worker $i$, the number of gradient steps taken will match the number of time slots $T$. Otherwise, the number of gradient steps taken will be $T \cdot p_i$ in expectation. MLL-SGD will wait $\tau$ time slots before averaging worker models in a sub-network, regardless of the number of gradient steps taken, while Local SGD and HL-SGD will wait for all workers to take $\tau$ gradient steps. This approach allows us to compare the progress of each algorithm over time. In this experiment, we set $p_i = 0.9$ for 90% of workers and $p_i = 0.6$ for 10% of the workers. As in the previous experiments, we use a multi-level network with a fully connected hub network and with 10 hubs, each with 10 workers. We study MLL-SGD with two parameter settings, $\tau = 32$, $q = 1$ and $\tau = 8$, $q = 4$. We also include results for Local-SGD and HL-SGD. By comparing MLL-SGD with $\tau = 32$, $q = 1$ with Local-SGD, we can evaluate the impact of using a local training period based on time rather than a number of worker iterations. By comparing MLL-SGD with $\tau = 8$, $q = 4$ with HL-SGD, we can evaluate this impact in a multi-level network.

In Figures 6a and 6c, we plot the training loss, and in Figures 6b and 6d, we plot the test accuracy for the CNN and ResNet, respectively. We can see that MLL-SGD with $q = 1$ converges more quickly, in both loss and accuracy, than Local SGD, and that MLL-SGD with $q = 4$ converges more quickly than HL-SGD. These trends hold in both the CNN and ResNet models. The results show that in this experimental setup, waiting for slow workers is detrimental to the overall convergence time.

## 7 CONCLUSION

We have introduced MLL-SGD, a variation of Distributed SGD in a multi-level network model. Our algorithm incorporates the heterogeneity of worker devices using a stochastic approach. We provide theoretical analysis of the algorithm's convergence, and we show how the convergence error depends on the average worker rate, the hub network topology, and the number of local, sub-network averaging, and hub averaging steps. Finally, we provide experimental results that illustrate the effectiveness of MLL-SGD over Local SGD and HL-SGD. In future work, we plan to analyze the effects of non-IID data on convergence error.

ACKNOWLEDGMENTS

This work is supported by the Rensselaer-IBM AI Research Collaboration (http://airc.rpi.edu), part of the IBM AI Horizons Network (http://ibm.biz/AIHorizons), and by the National Science Foundation under grants CNS 1553340 and CNS 1816307.

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

## A  Code Repository

The code used in our experiments can be found at: https://github.com/rpi-nsl/MLL-SGD. This code simulates a multi-level network with heterogeneous workers, and trains a model using MLL-SGD.

## B  Additional Experiments

Our experiments in Section 6 explore how changing MLL-SGD parameters affect training on a non-convex function. In this section, show the results of the same experiments on a convex loss function. We train a logistic regression model on the MNIST dataset (Bottou et al., 1994). We train a binary classification model with half the classes being 0 and the other half being 1 and use a step size of 0.2. We run all experiments for 32,000 iterations.

We rerun our first experiment from Figure 1 with logistic regression trained on MNIST. Figures 7a and 7b show the training loss and test accuracy, respectively. As with the non-convex functions, we can see that MLL-SGD with larger $q$ approaches the Distributed SGD baseline.

We rerun our second experiment comparing different hub and worker distributions with logistic regression trained on MNIST. Figure 8 shows the training loss. The three variations of MLL-SGD do not show much difference in terms of convergence rate, indicating that $\zeta$ has little effect in this case. However, they still outperform Local SGD due to $q$ being larger.

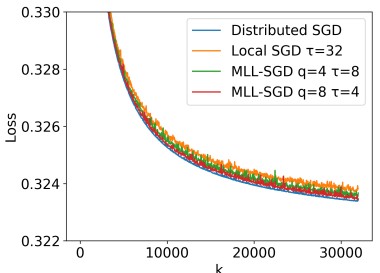

(a) Training loss of logistic regression trained on MNIST.

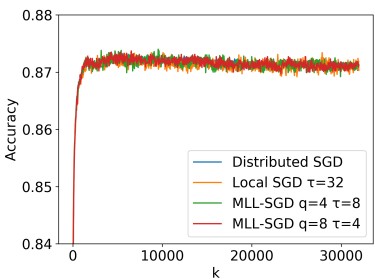

(b) Test accuracy of logistic regression trained on MNIST.

Figure 7: Effect of a hierarchy with different values of $\tau$ and $q$.

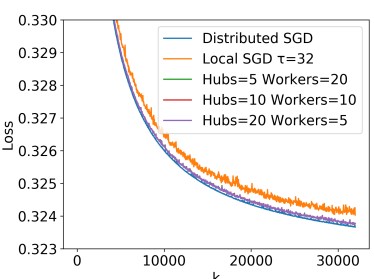

Figure 8: Effect of worker distribution on logistic regression trained on MNIST.

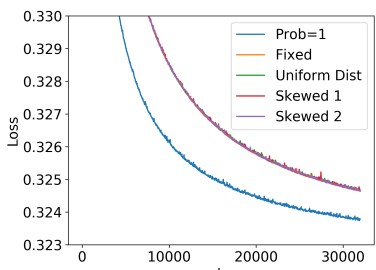

Figure 9: Effect of heterogeneous operating rates on logistic regression trained on MNIST.

We rerun our third experiment comparing different worker operating rates distributions with logistic regression trained on MNIST. Figure 9 shows the training loss. As with the non-convex functions, all MLL-SGD variations with the same average probability have similar convergence rate.

We rerun our first experiment from Figure 6 with logistic regression trained on MNIST. Figures 10a and 10b show the training loss and test accuracy, respectively. We can see an improvement in convergence rate of MLL-SGD over both Local SGD and HL-SGD.

## C  PROOF OF THEOREM 1

For our proof we adopt a similar approach to that in Wang & Joshi (2018). This section is structured as follows. We first define some notation and make some observations in Section C.1. Our supporting lemmas are stated in Section C.2. We close with the full proof of Theorem 1 in Section C.3.

### C.1  PRELIMINARIES

For simplicity of notation, we let $\|\cdot\|$ denote the $l_2$ vector norm. Let the weighted Frobenius norm of an $N \times M$ matrix $\boldsymbol{X}$ with an $N$-vector $\boldsymbol{a}$ be defined as follows:

$$\|\boldsymbol{X}\|_{F_{\boldsymbol{a}}}^2 = \left| \mathrm{Tr}((\mathrm{diag}(\boldsymbol{a}))^{1/2} \, \boldsymbol{X} \boldsymbol{X}^T \, (\mathrm{diag}(\boldsymbol{a}))^{1/2}) \right| = \sum_{i=1}^{N} \sum_{j=1}^{M} \boldsymbol{a}_i |\boldsymbol{x}_{i,j}|^2. \tag{16}$$

The matrix operator norm for a square matrix $\mathbf{Q}$ is defined as:

$$\|\mathbf{Q}\|_{op} = \sqrt{\lambda_{max}(\mathbf{Q}^T \mathbf{Q})}. \tag{17}$$

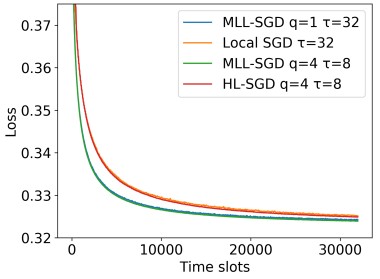

(a) Training loss of logistic regression with respect to time slots.

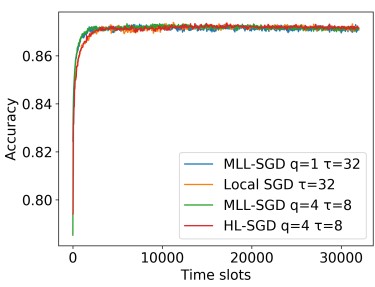

(b) Test accuracy of logistic regression with respect to time slots.

Figure 10: Comparing convergence time of Local SGD and MLL-SGD.

We define the set of Bernoulli random variables $\Theta = \{\theta_k^1, \dots, \theta_k^N\}$, where

$$\theta_k^i = \begin{cases} 1 & \text{with probability } \boldsymbol{p}_i \\ 0 & \text{with probability } (1 - \boldsymbol{p}_i). \end{cases}$$

Let $\Xi_k = \{\xi_k^{(1)}, \dots, \xi_k^{(N)}\}$ be the set of mini-batches used by the $N$ workers at time step $k$. Without loss of generality, we assign a mini-batch to each worker, even if it does not execute a gradient step in that iteration. An equivalent definition of $\boldsymbol{g}_k^{(i)}$ is then

$$\boldsymbol{g}_k^{(i)} = \theta_k^i g(\xi_k^{(i)}). \tag{18}$$

For simplicity of notation, let $\mathbb{E}_k$ be equivalent to $\mathbb{E}_{\Theta_k, \Xi_k | X_k}$.

We note that Assumption 1c implies:

$$\mathbb{E}_k[\boldsymbol{g}_k^{(i)}] = \boldsymbol{p}_i \mathbb{E}_k[g(\boldsymbol{x}_k^{(i)})] \tag{19}$$

$$= \boldsymbol{p}_i \nabla F(\boldsymbol{x}_k^{(i)}). \tag{20}$$

Further, when $i \neq j$:

$$\mathbb{E}_k[(\boldsymbol{g}_k^{(i)})^T \boldsymbol{g}_k^{(j)}] = \boldsymbol{p}_i \boldsymbol{p}_j \mathbb{E}_k[(g(\boldsymbol{x}_k^{(i)}))]^T \mathbb{E}_k[g(\boldsymbol{x}_k^{(j)})] \tag{21}$$

$$= \boldsymbol{p}_i \boldsymbol{p}_j \nabla F(\boldsymbol{x}_k^{(i)})^T \nabla F(\boldsymbol{x}_k^{(j)}). \tag{22}$$

We also note that Assumption 1d implies:

$$\mathbb{E}_k \left\| \boldsymbol{g}_k^{(i)} - \nabla F(\boldsymbol{x}_k^{(i)}) \right\|^2 = \mathbb{E}_k \left[ \left\| \boldsymbol{g}_k^{(i)} \right\|^2 + \left\| \nabla F(\boldsymbol{x}_k^{(i)}) \right\|^2 - 2(\boldsymbol{g}_k^{(i)})^T \nabla F(\boldsymbol{x}_k^{(i)}) \right] \tag{23}$$

$$= \mathbb{E}_k \left\| \boldsymbol{g}_k^{(i)} \right\|^2 + \left\| \nabla F(\boldsymbol{x}_k^{(i)}) \right\|^2 - 2\mathbb{E}_k(\boldsymbol{g}_k^{(i)})^T \nabla F(\boldsymbol{x}_k^{(i)}) \tag{24}$$

$$= \boldsymbol{p}_i \mathbb{E}_k \left\| g(\boldsymbol{x}_k^{(i)}) \right\|^2 + \left\| \nabla F(\boldsymbol{x}_k^{(i)}) \right\|^2 - 2\boldsymbol{p}_i \mathbb{E}_k g(\boldsymbol{x}_k^{(i)})^T \nabla F(\boldsymbol{x}_k^{(i)}) \tag{25}$$

$$= \boldsymbol{p}_i \mathbb{E}_k \left\| g(\boldsymbol{x}_k^{(i)}) \right\|^2 + \boldsymbol{p}_i \left\| \nabla F(\boldsymbol{x}_k^{(i)}) \right\|^2$$
$$\qquad - 2\boldsymbol{p}_i \mathbb{E}_k g(\boldsymbol{x}_k^{(i)})^T \nabla F(\boldsymbol{x}_k^{(i)}) + (1 - \boldsymbol{p}_i) \left\| \nabla F(\boldsymbol{x}_k^{(i)}) \right\|^2 \tag{26}$$

$$= \boldsymbol{p}_i \mathbb{E}_k \left\| g(\boldsymbol{x}_k^{(i)}) - \nabla F(\boldsymbol{x}_k^{(i)}) \right\|^2 + (1 - \boldsymbol{p}_i) \left\| \nabla F(\boldsymbol{x}_k^{(i)}) \right\|^2 \tag{27}$$

$$\leq \boldsymbol{p}_i \beta \left\| \nabla F(\boldsymbol{x}_k^{(i)}) \right\|^2 + \boldsymbol{p}_i \sigma^2 + (1 - \boldsymbol{p}_i) \left\| \nabla F(\boldsymbol{x}_k^{(i)}) \right\|^2 \tag{28}$$

$$= (\boldsymbol{p}_i(\beta - 1) + 1) \left\| \nabla F(\boldsymbol{x}_k^{(i)}) \right\|^2 + \boldsymbol{p}_i \sigma^2. \tag{29}$$

Finally, we define the weighted average stochastic gradient and the weighted average batch gradient as:

$$\mathcal{G}_k = \sum_{i=1}^{N} \boldsymbol{a}_i \boldsymbol{g}_k^{(i)}, \ \ \mathcal{H}_k = \sum_{i=1}^{N} \boldsymbol{a}_i \nabla F(\boldsymbol{x}_k^{(i)}).$$

## C.2  LEMMAS AND PROPOSITIONS

Next, we state our supporting lemmas and propositions.

**Proposition 1.** *The matrices $\boldsymbol{Z}$ and $\boldsymbol{V}$ satisfy the following properties:*

1. *$\boldsymbol{Z}$ and $\boldsymbol{V}$ each have a right eigenvector of $\boldsymbol{a}$ with eigenvalue 1.*

2. *$\boldsymbol{Z}$ and $\boldsymbol{V}$ each have a left eigenvector of $\boldsymbol{I}_N^T$ with eigenvalue 1.*

3. *All other eigenvalues of $\boldsymbol{Z}$ and $\boldsymbol{V}$ have magnitude strictly less than 1.*

*Proof.* Assumption 2 indicates that $\mathbf{H}$ is a *Generalized Diffusion Matrix* as defined in Rotaru & Nägeli (2004).

Recall Assumption 2:

**Assumption 2.** *The matrix $\mathbf{H}$ satisfies the following:*

2a *If $(i, j) \in E$, then $\mathbf{H}_{i,j} > 0$. Otherwise, $\mathbf{H}_{i,j} = 0$.*

2b *$\mathbf{H}$ is column stochastic, i.e., $\sum_{i=1}^{D} \mathbf{H}_{i,j} = 1$.*

2c *For all $i, j \in \mathcal{D}$, we have $\boldsymbol{b}_i \mathbf{H}_{i,j} = \boldsymbol{b}_j \mathbf{H}_{j,i}$.*

If we show this implies that $\boldsymbol{Z}$ and $\boldsymbol{V}$ are Generalized Diffusion Matrices with the same properties to those in Assumption 2 with vector $\boldsymbol{a}$, then the properties in the proposition are satisfied.

Since $\mathbf{H}$ and $\boldsymbol{b}$ are non-negative, then $\boldsymbol{Z}$ is also non-negative. It is also clear that $\boldsymbol{Z}$ is column stochastic by construction. It is left to prove that:

$$\boldsymbol{Z}_{i,j}\, \boldsymbol{a}_j = \boldsymbol{Z}_{j,i}\, \boldsymbol{a}_i. \tag{30}$$

Applying the definition of $\boldsymbol{Z}$ to the left side, we have:

$$\boldsymbol{Z}_{i,j}\, \boldsymbol{a}_j = \mathbf{H}_{d(i),d(j)} v^{(i)} \boldsymbol{a}_j \tag{31}$$

Since we know that $\mathbf{H}$ is a Generalized Diffusion Matrix with vector $\boldsymbol{b}$, we know that:

$$\mathbf{H}_{i,j}\boldsymbol{b}_j = \mathbf{H}_{j,i}\boldsymbol{b}_i \tag{32}$$

$$\mathbf{H}_{i,j} = \mathbf{H}_{j,i}\frac{\boldsymbol{b}_i}{\boldsymbol{b}_j}. \tag{33}$$

Plugging this in for $\mathbf{H}_{d(i),d(j)}$, we have:

$$\boldsymbol{Z}_{i,j}\, \boldsymbol{a}_j = \mathbf{H}_{d(j),d(i)}\frac{\boldsymbol{b}_{d(i)}}{\boldsymbol{b}_{d(j)}}v^{(i)}\boldsymbol{a}_j \tag{34}$$

$$= \mathbf{H}_{d(j),d(i)}\frac{\sum_{r\in\mathcal{M}^{(d(i))}} w^{(r)}}{w_{tot}}\frac{w_{tot}}{\sum_{r\in\mathcal{M}^{(d(j))}} w^{(r)}}\frac{w^{(i)}}{\sum_{r\in\mathcal{M}^{(d(i))}} w^{(r)}}\frac{w^{(j)}}{w_{tot}} \tag{35}$$

$$= \mathbf{H}_{d(j),d(i)}\frac{w^{(i)}}{w_{tot}}\frac{w^{(j)}}{\sum_{r\in\mathcal{M}^{(d(j))}} w^{(r)}} \tag{36}$$

$$= \mathbf{H}_{d(j),d(i)}v^{(j)}\boldsymbol{a}_i \tag{37}$$

$$= \boldsymbol{Z}_{j,i}\, \boldsymbol{a}_i. \tag{38}$$

Therefore, $\boldsymbol{Z}$ is a Generalized Diffusion Matrix.

We can show that $\mathbf{V}$ is also a Generalized Diffusion Matrix with the vector $\boldsymbol{a}$. $\mathbf{V}$ is constructed to be non-negative and column stochastic. It is left to prove that

$$\mathbf{V}_{i,j}\,\boldsymbol{a}_j = \mathbf{V}_{j,i}\,\boldsymbol{a}_i. \tag{39}$$

When $i, j$ are outside a block $\mathbf{V}^{(d)}$, then $\mathbf{V}_{i,j} = \mathbf{V}_{j,i} = 0$, so the equation is trivially satisfied. When within a block, in terms of $w$, we have:

$$\mathbf{V}_{i,j}\,\boldsymbol{a}_j = \mathbf{V}_{j,i}\,\boldsymbol{a}_i \tag{40}$$

$$\frac{w^{(i)}}{\sum_{r \in \mathcal{M}^{(d(i))}} w^{(r)}} \frac{w^{(j)}}{w_{tot}} = \frac{w^{(j)}}{\sum_{r \in \mathcal{M}^{(d(j))}} w^{(r)}} \frac{w^{(i)}}{w_{tot}}. \tag{41}$$

Noting that we are within a block, therefore $d(i) = d(j)$, we can see that both sides are equal:

$$w^{(i)}w^{(j)} = w^{(j)}w^{(i)}. \tag{42}$$

Therefore, $\mathbf{V}$ is a Generalized Diffusion Matrix.

$\square$

**Proposition 2.** *Given a diffusion matrix $\mathbf{H}$ with the properties in Assumption 2, if $\mathbf{Z}$ constructed as follows,*

$$\mathbf{Z}_{i,j} = \mathbf{H}_{d(i),d(j)}v^{(i)} \tag{43}$$

*then the largest eigenvalues of $\mathbf{Z}$ are the eigenvalues of $\mathbf{H}$, and zero otherwise.*

*Proof.* In order to prove the relationship of the eigenvalues of $\mathbf{Z}$ and $\mathbf{H}$, we prove the following two points separately:

1. The rank of $\mathbf{Z}$ is the same as $\mathbf{H}$.

2. All non-zero eigenvalues of $\mathbf{H}$ are eigenvalues of $\mathbf{Z}$ with the same multiplicity.

For the rank of $\mathbf{Z}$, we take a look at how each column is constructed. Consider column $j$ of $\mathbf{Z}$:

$$\mathbf{Z}_j = [\mathbf{H}_{1,d(j)}v^{(1)}, \ldots, \mathbf{H}_{1,d(j)}v^{(N^{(1)})}, \mathbf{H}_{2,d(j)}v^{(N^{(1)}+1)}, \ldots, \mathbf{H}_{D,d(j)}v^{(N)}]^T. \tag{44}$$

For two columns $i$ and $j$ where $d(i) = d(j)$, these columns are identical. Therefore, the rank of $\mathbf{Z}$ will be, at most, the number of hubs, $D$. Further, we can see that the elements of a column $j$ in $\mathbf{Z}$ are simply scaled elements of column $d(j)$ in $\mathbf{H}$. So any linearly dependent columns in $\mathbf{H}$ will also be linearly dependent in $\mathbf{Z}$. Therefore, the rank of the two matrices are the same.

For the second point, we show there is a bijective mapping from eigenpairs of $\mathbf{H}$ to eigenpairs of $\mathbf{Z}$. Let $(\lambda, \boldsymbol{y})$ be an eigenpair of $\mathbf{H}$ (with $\lambda \neq 0$), i.e.

$$\mathbf{H}\boldsymbol{y} = \lambda\boldsymbol{y}. \tag{45}$$

Define the $N$-vector $\boldsymbol{x}$ with components $\boldsymbol{x}_i = v^{(i)}\boldsymbol{y}_{d(i)}$. We will show that $\mathbf{Z}\,\boldsymbol{x} = \lambda\boldsymbol{x}$. Looking at the $i$-th entry of the vector $\mathbf{Z}\,\boldsymbol{x}$, we have

$$(\mathbf{Z}\,\boldsymbol{x})_i = \sum_{j=1}^{N} \mathbf{Z}_{i,j}\,\boldsymbol{x}_j. \tag{46}$$

Applying the definition of $\mathbf{Z}$ and $\boldsymbol{x}$, we obtain

$$(\mathbf{Z}\,\boldsymbol{x})_i = \sum_{j=1}^{N} \frac{1}{v^{(i)}}\mathbf{H}_{d(i),d(j)}v^{(j)}\boldsymbol{y}_{d(j)} \tag{47}$$

$$= \frac{1}{v^{(i)}}\sum_{l=1}^{D}\mathbf{H}_{d(i),l}\boldsymbol{y}_l \sum_{k \in \mathcal{M}^{(l)}} v^{(k)} \tag{48}$$

$$= \frac{1}{v^{(i)}}\sum_{l=1}^{D}\mathbf{H}_{d(i),l}\boldsymbol{y}_l. \tag{49}$$

Note that the $m$-th entry of the vector $\mathbf{H}\boldsymbol{y}$ equals $\sum_{l=1}^{D} \mathbf{H}_{m,l}\boldsymbol{y}_l = \lambda\boldsymbol{y}_m$. Applying this equality, we obtain

$$(\mathbf{Z}\boldsymbol{x})_i = \frac{1}{v^{(i)}}\lambda\boldsymbol{y}_{d(i)} \tag{50}$$

$$= \lambda\boldsymbol{x}_i. \tag{51}$$

Therefore, for any eigenpair $(\lambda, \boldsymbol{y})$ of $\mathbf{H}$, we can find an eigenpair $(\lambda, \boldsymbol{x})$ of $\mathbf{Z}$. It is left to prove that this mapping is a bijection.

Suppose eigenvalue $\lambda$ of $\mathbf{H}$ has multiplicity $k > 1$. We consider any two of the $k$ eigenpairs $(\lambda, \boldsymbol{c})$ and $(\lambda, \boldsymbol{d})$. Let the corresponding eigenpairs of $\mathbf{Z}$ be $(\lambda, \boldsymbol{e})$ and $(\lambda, \boldsymbol{f})$. We know that $\boldsymbol{e} \neq \boldsymbol{f}$ because $\boldsymbol{c}$ and $\boldsymbol{d}$ are unique, and there must exist an index $i$ such that $v^{(i)}\boldsymbol{c}_{d(i)} \neq v^{(i)}\boldsymbol{d}_{d(i)}$. Therefore, the mapping of eigenpairs of $\mathbf{H}$ to eigenpairs of $\mathbf{Z}$ is a bijection. $\qquad\square$

**Proposition 3.** *Given definition of $\mathbf{Z}$ and $\mathbf{V}$ in Proposition 1, it is the case that*

$$\mathbf{Z}\mathbf{V} = \mathbf{V}\mathbf{Z} = \mathbf{Z}. \tag{52}$$

*Proof.* First, we prove that $\mathbf{V}\mathbf{Z} = \mathbf{Z}$. Note that the $i$-th row of $\mathbf{V}$ contains either $v_i$ or zero. Looking at an arbitrary entry $i, j$ of $\mathbf{V}\mathbf{Z}$ we have:

$$(\mathbf{V}\mathbf{Z})_{i,j} = v^{(i)} \sum_{r \in M_{d(i)}} \mathbf{Z}_{r,j} \tag{53}$$

$$(\mathbf{V}\mathbf{Z})_{i,j} = v^{(i)}\mathbf{H}_{d(i),d(i)} \tag{54}$$

$$(\mathbf{V}\mathbf{Z})_{i,j} = \mathbf{Z}_{i,j}. \tag{55}$$

Next we prove that $\mathbf{Z}\mathbf{V} = \mathbf{Z}$. Note that for any row $i$ in $\mathbf{Z}$, $\mathbf{Z}_{i,j} = \mathbf{Z}_{i,k}$ when $d(j) = d(k)$.

$$(\mathbf{Z}\mathbf{V})_{i,j} = \mathbf{Z}_{i,j} \sum_{r=1}^{N} \mathbf{V}_{r,j}. \tag{56}$$

Since $\mathbf{V}$ is column stochastic:

$$(\mathbf{Z}\mathbf{V})_{i,j} = \mathbf{Z}_{i,j}. \tag{57}$$

$$\square$$

**Proposition 4.** *Let $\mathbf{A} = \boldsymbol{a}\mathbf{1}^T$. Given our definition of $\mathbf{T}_k$ in (6),*

$$\mathbf{T}_k\mathbf{A} = \mathbf{A}\mathbf{T}_k = \mathbf{A} \tag{58}$$

*for all $k$.*

*Proof.* We prove each of the three cases of $\mathbf{T}_k$: $\mathbf{I}$, $\mathbf{V}$, and $\mathbf{Z}$. Clearly, $\mathbf{I}\mathbf{A} = \mathbf{A}\mathbf{I} = \mathbf{A}$. It is left to prove $\mathbf{V}\mathbf{A} = \mathbf{A}\mathbf{V} = \mathbf{A}$ and $\mathbf{Z}\mathbf{A} = \mathbf{A}\mathbf{Z} = \mathbf{A}$.

We can see that $\mathbf{Z}\mathbf{A} = \mathbf{A}$ since $\boldsymbol{a}$ is a right eigenvector of $\mathbf{Z}$ with eigenvalue 1: $\mathbf{Z}\mathbf{A} = \mathbf{Z}\boldsymbol{a}\mathbf{1}^T = \boldsymbol{a}\mathbf{1}^T = \mathbf{A}$. Similarly, we can see that $\mathbf{A}\mathbf{Z} = \boldsymbol{a}\mathbf{1}^T\mathbf{Z} = \boldsymbol{a}\mathbf{1}^T = \mathbf{A}$ as $\mathbf{1}^T$ is a left eigenvector of $\mathbf{Z}$. The same holds for $\mathbf{V}$. $\qquad\square$

**Lemma 1.** *Under Assumptions 1c and 1d, the variance of the weighted average stochastic gradient is bounded as follows:*

$$\mathbb{E}_k[\|\mathcal{G}_k - \mathcal{H}_k\|^2] \leq \sum_{i=1}^{N} \boldsymbol{a}_i^2 \left[ (\boldsymbol{p}_i(\beta - 1) + 1)\left\|\nabla F(\boldsymbol{x}_k^{(i)})\right\|^2 + \boldsymbol{p}_i\sigma^2 \right]$$

$$+ \sum_{l=1}^{N}\sum_{j \neq l}^{N} \boldsymbol{a}_l\boldsymbol{a}_j(1 - \boldsymbol{p}_j)(1 - \boldsymbol{p}_l)\nabla F(\boldsymbol{x}_k^{(j)})^T\nabla F(\boldsymbol{x}_k^{(l)}). \tag{59}$$

*Proof.*

$$\mathbb{E}_k[\|\mathcal{G}_k - \mathcal{H}_k\|^2] = \mathbb{E}_k\left[\left\|\sum_{i=1}^{N} \boldsymbol{a}_i(\boldsymbol{g}_k^{(i)} - \nabla F(\boldsymbol{x}_k))\right\|^2\right] \tag{60}$$

$$= \mathbb{E}_k\left[\sum_{i=1}^{N} \boldsymbol{a}_i^2\|\boldsymbol{g}_k^{(i)} - \nabla F(\boldsymbol{x}_k)\|^2 + \sum_{l=1}^{N}\sum_{j\neq l}^{N} \boldsymbol{a}_l\boldsymbol{a}_j\left\langle\boldsymbol{g}_k^{(j)} - \nabla F(\boldsymbol{x}_k^{(j)}), \boldsymbol{g}_k^{(l)} - \nabla F(\boldsymbol{x}_k^{(l)})\right\rangle\right] \tag{61}$$

$$= \sum_{i=1}^{N} \boldsymbol{a}_i^2\mathbb{E}_k\|\boldsymbol{g}_k^{(i)} - \nabla F(\boldsymbol{x}_k)\|^2 + \sum_{l=1}^{N}\sum_{j\neq l}^{N} \boldsymbol{a}_l\boldsymbol{a}_j\mathbb{E}_k\left[\left\langle\boldsymbol{g}_k^{(j)} - \nabla F(\boldsymbol{x}_k^{(j)}), \boldsymbol{g}_k^{(l)} - \nabla F(\boldsymbol{x}_k^{(l)})\right\rangle\right]. \tag{62}$$

Looking at the cross-terms in (62):

$$\mathbb{E}_k\left[\left\langle\boldsymbol{g}_k^{(j)} - \nabla F(\boldsymbol{x}_k^{(j)}), \boldsymbol{g}_k^{(l)} - \nabla F(\boldsymbol{x}_k^{(l)})\right\rangle\right]$$

$$= \mathbb{E}_k\left[(\boldsymbol{g}_k^{(j)})^T\boldsymbol{g}_k^{(l)}\right] - \mathbb{E}_k\left[(\boldsymbol{g}_k^{(j)})^T\nabla F(\boldsymbol{x}_k^{(l)})\right]$$
$$- \mathbb{E}_k\left[\nabla F(\boldsymbol{x}_k^{(j)})^T\boldsymbol{g}_k^{(l)}\right] + \nabla F(\boldsymbol{x}_k^{(j)})^T\nabla F(\boldsymbol{x}_k^{(j)}) \tag{63}$$

$$= \boldsymbol{p}_j\boldsymbol{p}_l\nabla F(\boldsymbol{x}_k^{(j)})^T\nabla F(\boldsymbol{x}_k^{(l)}) - \boldsymbol{p}_j\nabla F(\boldsymbol{x}_k^{(j)})^T\nabla F(\boldsymbol{x}_k^{(l)})$$
$$- \boldsymbol{p}_l\nabla F(\boldsymbol{x}_k^{(j)})^T\nabla F(\boldsymbol{x}_k^{(l)}) + \nabla F(\boldsymbol{x}_k^{(j)})^T\nabla F(\boldsymbol{x}_k^{(j)}) \tag{64}$$

$$= (1 - \boldsymbol{p}_j)(1 - \boldsymbol{p}_l)\nabla F(\boldsymbol{x}_k^{(j)})^T\nabla F(\boldsymbol{x}_k^{(l)}). \tag{65}$$

Plugging (65) into (62) we have:

$$\mathbb{E}_k[\|\mathcal{G}_k - \mathcal{H}_k\|^2] = \sum_{i=1}^{N} \boldsymbol{a}_i^2\mathbb{E}_k\|\boldsymbol{g}_k^{(i)} - \nabla F(\boldsymbol{x}_k)\|^2$$

$$+ \sum_{l=1}^{N}\sum_{j\neq l}^{N} \boldsymbol{a}_l\boldsymbol{a}_j(1 - \boldsymbol{p}_j)(1 - \boldsymbol{p}_l)\nabla F(\boldsymbol{x}_k^{(j)})^T\nabla F(\boldsymbol{x}_k^{(l)}) \tag{66}$$

$$\leq \sum_{i=1}^{N} \boldsymbol{a}_i^2\left[(\boldsymbol{p}_i(\beta - 1) + 1)\left\|\nabla F(\boldsymbol{x}_k^{(i)})\right\|^2 + \boldsymbol{p}_i\sigma^2\right]$$

$$+ \sum_{l=1}^{N}\sum_{j\neq l}^{N} \boldsymbol{a}_l\boldsymbol{a}_j(1 - \boldsymbol{p}_j)(1 - \boldsymbol{p}_l)\nabla F(\boldsymbol{x}_k^{(j)})^T\nabla F(\boldsymbol{x}_k^{(l)}) \tag{67}$$

where (67) follows from Assumption 1d and (29). $\qquad\square$

**Lemma 2.** *Under Assumptions 1c and 1d, the squared norm of the stochastic gradients is bounded by:*

$$\mathbb{E}_k\left[\|\mathcal{G}_k\|^2\right] \leq \sum_{i=1}^{N}\left[\boldsymbol{a}_i^2\left(\boldsymbol{p}_i(\beta + 1) - \boldsymbol{p}_i^2\right) + \boldsymbol{a}_i\boldsymbol{p}_i^2\right]\left\|\nabla F(\boldsymbol{x}_k^{(i)})\right\|^2 + \sum_{i=1}^{N} \boldsymbol{a}_i^2\boldsymbol{p}_i\sigma^2 \tag{68}$$

*Proof.*

$$\mathbb{E}_k\left[\|\mathcal{G}_k\|^2\right] = \mathbb{E}_k\left[\|\mathcal{G}_k - \mathbb{E}_k[\mathcal{G}_k]\|^2\right] + \|\mathbb{E}_k[\mathcal{G}_k]\|^2 \tag{69}$$

$$= \mathbb{E}_k\left[\|\mathcal{G}_k - \sum_{i=1}^{N} \boldsymbol{a}_i\boldsymbol{p}_i\nabla F(\boldsymbol{x}_k^{(i)})\|^2\right] + \left\|\sum_{i=1}^{N} \boldsymbol{a}_i\boldsymbol{p}_i\nabla F(\boldsymbol{x}_k^{(i)})\right\|^2 \tag{70}$$

$$= \mathbb{E}_k\left[\left\|\mathcal{G}_k - \sum_{i=1}^{N} \boldsymbol{a}_i\nabla F(\boldsymbol{x}_k^{(i)}) + \sum_{i=1}^{N} \boldsymbol{a}_i(1 - \boldsymbol{p}_i)\nabla F(\boldsymbol{x}_k^{(i)})\right\|^2\right] + \left\|\sum_{i=1}^{N} \boldsymbol{a}_i\boldsymbol{p}_i\nabla F(\boldsymbol{x}_k^{(i)})\right\|^2. \tag{71}$$

Applying the definition of $\mathcal{G}_k$ to (71) we get:

$$\mathbb{E}_k\left[\|\mathcal{G}_k\|^2\right]$$

$$= \mathbb{E}_k\left[\left\|\sum_{i=1}^{N}\left[\boldsymbol{a}_i\boldsymbol{g}_k^{(i)} - \boldsymbol{a}_i\nabla F(\boldsymbol{x}_k^{(i)}) + \boldsymbol{a}_i(1-\boldsymbol{p}_i)\nabla F(\boldsymbol{x}_k^{(i)})\right]\right\|^2\right] + \left\|\sum_{i=1}^{N}\boldsymbol{a}_i\boldsymbol{p}_i\nabla F(\boldsymbol{x}_k^{(i)})\right\|^2 \quad (72)$$

$$= \mathbb{E}_k\left[\sum_{i=1}^{N}\left\|\boldsymbol{a}_i\boldsymbol{g}_k^{(i)} - \boldsymbol{a}_i\nabla F(\boldsymbol{x}_k^{(i)}) + \boldsymbol{a}_i(1-\boldsymbol{p}_i)\nabla F(\boldsymbol{x}_k^{(i)})\right\|^2\right] + \left\|\sum_{i=1}^{N}\boldsymbol{a}_i\boldsymbol{p}_i\nabla F(\boldsymbol{x}_k^{(i)})\right\|^2$$

$$+ \mathbb{E}_k\left[\sum_{j=1}^{N}\sum_{l=1,l\neq j}^{N}\boldsymbol{a}_l\boldsymbol{a}_j\left\langle(\boldsymbol{g}_k^{(j)} - \nabla F(\boldsymbol{x}_k^{(j)})) + (1-\boldsymbol{p}_j)\nabla F(\boldsymbol{x}_k^{(j)}),\right.$$

$$\left.(\boldsymbol{g}_k^{(l)} - \nabla F(\boldsymbol{x}_k^{(l)})) + (1-\boldsymbol{p}_l)\nabla F(\boldsymbol{x}_k^{(l)})\right\rangle\right]. \quad (73)$$

Let the cross-terms in (73) be

$$CR = \mathbb{E}_k\left[\sum_{j=1}^{N}\sum_{l=1,l\neq j}^{N}\boldsymbol{a}_l\boldsymbol{a}_j\left\langle(\boldsymbol{g}_k^{(j)} - \nabla F(\boldsymbol{x}_k^{(j)})) + (1-\boldsymbol{p}_j)\nabla F(\boldsymbol{x}_k^{(j)}),\right.\right.$$

$$\left.\left.(\boldsymbol{g}_k^{(l)} - \nabla F(\boldsymbol{x}_k^{(l)})) + (1-\boldsymbol{p}_l)\nabla F(\boldsymbol{x}_k^{(l)})\right\rangle\right]. \quad (74)$$

We can simplify $CR$ as follows:

$$CR = \sum_{j=1}^{N}\sum_{l=1,l\neq j}^{N}\boldsymbol{a}_l\boldsymbol{a}_j\mathbb{E}_k\left[(\boldsymbol{g}_k^{(j)} - \nabla F(\boldsymbol{x}_k^{(j)}))^T(\boldsymbol{g}_k^{(l)} - \nabla F(\boldsymbol{x}_k^{(l)}))\right.$$

$$+ (\boldsymbol{g}_k^{(j)} - \nabla F(\boldsymbol{x}_k^{(j)}))^T(1-\boldsymbol{p}_l)\nabla F(\boldsymbol{x}_k^{(l)}) + (1-\boldsymbol{p}_j)\nabla F(\boldsymbol{x}_k^{(j)})^T(\boldsymbol{g}_k^{(l)} - \nabla F(\boldsymbol{x}_k^{(l)}))$$

$$\left. + (1-\boldsymbol{p}_j)(1-\boldsymbol{p}_l)\nabla F(\boldsymbol{x}_k^{(j)})^T\nabla F(\boldsymbol{x}_k^{(l)})^T\right] \quad (75)$$

$$= \sum_{j=1}^{N}\sum_{l=1,l\neq j}^{N}\boldsymbol{a}_l\boldsymbol{a}_j\left[\mathbb{E}_k\left[(\boldsymbol{g}_k^{(j)} - \nabla F(\boldsymbol{x}_k^{(j)}))^T(\boldsymbol{g}_k^{(l)} - \nabla F(\boldsymbol{x}_k^{(l)}))\right]\right.$$

$$+ (\mathbb{E}_k[\boldsymbol{g}_k^{(j)}] - \nabla F(\boldsymbol{x}_k^{(j)}))^T(1-\boldsymbol{p}_l)\nabla F(\boldsymbol{x}_k^{(l)})$$

$$+ (1-\boldsymbol{p}_j)\nabla F(\boldsymbol{x}_k^{(j)})^T(\mathbb{E}_k[\boldsymbol{g}_k^{(l)}] - \nabla F(\boldsymbol{x}_k^{(l)}))$$

$$\left. + (1-\boldsymbol{p}_j)(1-\boldsymbol{p}_l)\nabla F(\boldsymbol{x}_k^{(j)})^T\nabla F(\boldsymbol{x}_k^{(l)})^T\right]. \quad (76)$$

Applying Assumption 1c to (76), we get:

$$CR = \sum_{j=1}^{N}\sum_{l=1,l\neq j}^{N}\boldsymbol{a}_l\boldsymbol{a}_j\left[\mathbb{E}_k\left[(\boldsymbol{g}_k^{(j)} - \nabla F(\boldsymbol{x}_k^{(j)}))^T(\boldsymbol{g}_k^{(l)} - \nabla F(\boldsymbol{x}_k^{(l)}))\right]\right.$$

$$+ (\boldsymbol{p}_j - 1)(1-\boldsymbol{p}_l)\nabla F(\boldsymbol{x}_k^{(j)})^T\nabla F(\boldsymbol{x}_k^{(l)}) + (\boldsymbol{p}_l - 1)(1-\boldsymbol{p}_j)\nabla F(\boldsymbol{x}_k^{(j)})^T\nabla F(\boldsymbol{x}_k^{(l)})$$

$$\left. + (1-\boldsymbol{p}_j)(1-\boldsymbol{p}_l)\nabla F(\boldsymbol{x}_k^{(j)})^T\nabla F(\boldsymbol{x}_k^{(l)})\right] \quad (77)$$

$$= \sum_{j=1}^{N}\sum_{l=1,l\neq j}^{N}\boldsymbol{a}_l\boldsymbol{a}_j\left[\mathbb{E}_k\left[(\boldsymbol{g}_k^{(j)} - \nabla F(\boldsymbol{x}_k^{(j)}))^T(\boldsymbol{g}_k^{(l)} - \nabla F(\boldsymbol{x}_k^{(l)}))\right]\right.$$

$$\left. - (1-\boldsymbol{p}_l)(1-\boldsymbol{p}_j)\nabla F(\boldsymbol{x}_k^{(j)})^T\nabla F(\boldsymbol{x}_k^{(l)})\right]. \quad (78)$$

Applying (78) to (73) we have:

$$
\mathbb{E}_k\left[\|\mathcal{G}_k\|^2\right] = \mathbb{E}_k\left[\sum_{i=1}^N \left\|\boldsymbol{a}_i\boldsymbol{g}_k^{(i)} - \boldsymbol{a}_i\nabla F(\boldsymbol{x}_k^{(i)}) + \boldsymbol{a}_i(1-\boldsymbol{p}_i)\nabla F(\boldsymbol{x}_k^{(i)})\right\|^2\right]
$$
$$
+ \sum_{j=1}^N\sum_{l=1,l\neq j}^N \boldsymbol{a}_l\boldsymbol{a}_j\left[\mathbb{E}_k\left[(\boldsymbol{g}_k^{(j)} - \nabla F(\boldsymbol{x}_k^{(j)}))^T(\boldsymbol{g}_k^{(l)} - \nabla F(\boldsymbol{x}_k^{(l)}))\right]\right.
$$
$$
\left. - (1-\boldsymbol{p}_l)(1-\boldsymbol{p}_j)\nabla F(\boldsymbol{x}_k^{(j)})^T\nabla F(\boldsymbol{x}_k^{(l)})\right]
$$
$$
+ \left\|\sum_{i=1}^N \boldsymbol{a}_i\boldsymbol{p}_i\nabla F(\boldsymbol{x}_k^{(i)})\right\|^2. \tag{79}
$$

Expanding the first term in (79) we have:

$$
\mathbb{E}_k\left[\|\mathcal{G}_k\|^2\right] = \mathbb{E}_k\left[\sum_{i=1}^N \left\|\boldsymbol{a}_i\boldsymbol{g}_k^{(i)} - \boldsymbol{a}_i\nabla F(\boldsymbol{x}_k^{(i)})\right\|^2\right] + \sum_{i=1}^N \boldsymbol{a}_i^2(1-\boldsymbol{p}_i)^2\left\|\nabla F(\boldsymbol{x}_k^{(i)})\right\|^2
$$
$$
+ \sum_{i=1}^N \underbrace{\mathbb{E}_k\left[\left\langle \boldsymbol{a}_i\boldsymbol{g}_k^{(i)} - \boldsymbol{a}_i\nabla F(\boldsymbol{x}_k^{(i)}), \boldsymbol{a}_i(1-\boldsymbol{p}_i)\nabla F(\boldsymbol{x}_k^{(i)})\right\rangle\right]}_{CR1}
$$
$$
+ \sum_{i=1}^N \underbrace{\mathbb{E}_k\left[\left\langle \boldsymbol{a}_i(1-\boldsymbol{p}_i)\nabla F(\boldsymbol{x}_k^{(i)}), \boldsymbol{a}_i\boldsymbol{g}_k^{(i)} - \boldsymbol{a}_i\nabla F(\boldsymbol{x}_k^{(i)})\right\rangle\right]}_{CR2}
$$
$$
+ \sum_{j=1}^N\sum_{l=1,l\neq j}^N \boldsymbol{a}_l\boldsymbol{a}_j\left[\mathbb{E}_k\left[(\boldsymbol{g}_k^{(j)} - \nabla F(\boldsymbol{x}_k^{(j)}))^T(\boldsymbol{g}_k^{(l)} - \nabla F(\boldsymbol{x}_k^{(l)}))\right]\right.
$$
$$
\left. - (1-\boldsymbol{p}_l)(1-\boldsymbol{p}_j)\nabla F(\boldsymbol{x}_k^{(j)})^T\nabla F(\boldsymbol{x}_k^{(l)})\right]
$$
$$
+ \left\|\sum_{i=1}^N \boldsymbol{a}_i\boldsymbol{p}_i\nabla F(\boldsymbol{x}_k^{(i)})\right\|^2. \tag{80}
$$

We simplify $CR1$:

$$
CR1 = \left[\left\langle \boldsymbol{a}_i\boldsymbol{p}_i\mathbb{E}_k(g(\boldsymbol{x}_k^{(i)}))^T - \boldsymbol{a}_i\nabla F(\boldsymbol{x}_k^{(i)}), \boldsymbol{a}_i(1-\boldsymbol{p}_i)\nabla F(\boldsymbol{x}_k^{(i)})\right\rangle\right] \tag{81}
$$
$$
= \left[\left\langle \boldsymbol{a}_i(\boldsymbol{p}_i-1)\nabla F(\boldsymbol{x}_k^{(i)}), \boldsymbol{a}_i(1-\boldsymbol{p}_i)\nabla F(\boldsymbol{x}_k^{(i)})\right\rangle\right] \tag{82}
$$
$$
= -\boldsymbol{a}_i^2(1-\boldsymbol{p}_i)^2\left\|\nabla F(\boldsymbol{x}_k^{(i)})\right\|^2. \tag{83}
$$

Similarly, for $CR2$:

$$
CR2 = -\boldsymbol{a}_i^2(1-\boldsymbol{p}_i)^2\left\|\nabla F(\boldsymbol{x}_k^{(i)})\right\|^2. \tag{84}
$$

Plugging (83) and (84) back into (80):

$$
\begin{aligned}
\mathbb{E}_k\left[\|\mathcal{G}_k\|^2\right] = \mathbb{E}_k\Bigg[\sum_{i=1}^N\left\|\boldsymbol{a}_i\boldsymbol{g}_k^{(i)} - \boldsymbol{a}_i\nabla F(\boldsymbol{x}_k^{(i)})\right\|^2\Bigg] &- \sum_{i=1}^N\boldsymbol{a}_i^2(1-\boldsymbol{p}_i)^2\left\|\nabla F(\boldsymbol{x}_k^{(i)})\right\|^2 \\
+ \sum_{j=1}^N\sum_{l=1,l\neq j}^N \boldsymbol{a}_l\boldsymbol{a}_j\Big[\mathbb{E}_k\Big[(\boldsymbol{g}_k^{(j)} &- \nabla F(\boldsymbol{x}_k^{(j)}))^T(\boldsymbol{g}_k^{(l)} - \nabla F(\boldsymbol{x}_k^{(l)}))\Big] \\
&- (1-\boldsymbol{p}_l)(1-\boldsymbol{p}_j)\nabla F(\boldsymbol{x}_k^{(j)})^T\nabla F(\boldsymbol{x}_k^{(l)})\Big] \\
&+ \left\|\sum_{i=1}^N\boldsymbol{a}_i\boldsymbol{p}_i\nabla F(\boldsymbol{x}_k^{(i)})\right\|^2. \quad (85)
\end{aligned}
$$

We can simplify by observing that:

$$
\begin{aligned}
\mathbb{E}_k\left[\|\mathcal{G}_k - \mathcal{H}_k\|^2\right] = \mathbb{E}_k\Bigg[\sum_{i=1}^N\left\|\boldsymbol{a}_i\boldsymbol{g}_k^{(i)} - \boldsymbol{a}_i\nabla F(\boldsymbol{x}_k^{(i)})\right\|^2\Bigg] \\
+ \sum_{j=1}^N\sum_{l=1,l\neq j}^N \boldsymbol{a}_l\boldsymbol{a}_j\mathbb{E}_k\Big[(\boldsymbol{g}_k^{(j)} - \nabla F(\boldsymbol{x}_k^{(j)}))^T(\boldsymbol{g}_k^{(l)} - \nabla F(\boldsymbol{x}_k^{(l)}))\Big] \quad (86)
\end{aligned}
$$

which gives us:

$$
\begin{aligned}
\mathbb{E}_k\left[\|\mathcal{G}_k\|^2\right] = \mathbb{E}_k\left[\|\mathcal{G}_k - \mathcal{H}_k\|^2\right] - \sum_{i=1}^N\boldsymbol{a}_i^2(1-\boldsymbol{p}_i)^2\left\|\nabla F(\boldsymbol{x}_k^{(i)})\right\|^2 + \left\|\sum_{i=1}^N\boldsymbol{a}_i\boldsymbol{p}_i\nabla F(\boldsymbol{x}_k^{(i)})\right\|^2 \\
- \sum_{j=1}^N\sum_{l=1,l\neq j}^N \boldsymbol{a}_l\boldsymbol{a}_j(1-\boldsymbol{p}_l)(1-\boldsymbol{p}_j)\nabla F(\boldsymbol{x}_k^{(j)})^T\nabla F(\boldsymbol{x}_k^{(l)}) \quad (87)
\end{aligned}
$$

Applying Lemma 1 to (87):

$$
\begin{aligned}
\mathbb{E}_k\left[\|\mathcal{G}_k\|^2\right] \leq \sum_{i=1}^N\boldsymbol{a}_i^2\left[(\boldsymbol{p}_i(\beta-1)+1)\left\|\nabla F(\boldsymbol{x}_k^{(i)})\right\|^2 + \boldsymbol{p}_i\sigma^2\right] \\
- \sum_{i=1}^N\boldsymbol{a}_i^2(1-\boldsymbol{p}_i)^2\left\|\nabla F(\boldsymbol{x}_k^{(i)})\right\|^2 + \left\|\sum_{i=1}^N\boldsymbol{a}_i\boldsymbol{p}_i\nabla F(\boldsymbol{x}_k^{(i)})\right\|^2 \quad (88)
\end{aligned}
$$

$$
\begin{aligned}
\leq \sum_{i=1}^N\boldsymbol{a}_i^2\left[(\boldsymbol{p}_i(\beta-1)+1)\left\|\nabla F(\boldsymbol{x}_k^{(i)})\right\|^2 + \boldsymbol{p}_i\sigma^2\right] \\
- \sum_{i=1}^N\boldsymbol{a}_i^2(1-\boldsymbol{p}_i)^2\left\|\nabla F(\boldsymbol{x}_k^{(i)})\right\|^2 + \sum_{i=1}^N\boldsymbol{a}_i\boldsymbol{p}_i^2\left\|\nabla F(\boldsymbol{x}_k^{(i)})\right\|^2 \quad (89)
\end{aligned}
$$

$$
= \sum_{i=1}^N\left[\boldsymbol{a}_i^2\left(\boldsymbol{p}_i(\beta+1)-\boldsymbol{p}_i^2\right)+\boldsymbol{a}_i\boldsymbol{p}_i^2\right]\left\|\nabla F(\boldsymbol{x}_k^{(i)})\right\|^2 + \sum_{i=1}^N\boldsymbol{a}_i^2\boldsymbol{p}_i\sigma^2 \quad (90)
$$

where equation (89) follows from Jensen's inequality. $\square$

**Lemma 3.** *Under Assumption 1c, the expected inner product of the batch gradient and the weighted average stochastic gradient is equal to:*

$$
\begin{aligned}
\mathbb{E}_k[\langle\nabla F(\boldsymbol{u}_k), \mathcal{G}_k\rangle] = \frac{1}{2}\|\nabla F(\boldsymbol{u}_k)\|^2 + \sum_{i=1}^N\frac{\boldsymbol{a}_i}{2}\left\|\boldsymbol{p}_i\nabla F(\boldsymbol{x}_k^{(i)})\right\|^2 \\
- \sum_{i=1}^N\frac{\boldsymbol{a}_i}{2}\left\|\nabla F(\boldsymbol{u}_k) - \boldsymbol{p}_i\nabla F(\boldsymbol{x}_k^{(i)})\right\|^2 \quad (91)
\end{aligned}
$$

*Proof.*

$$\mathbb{E}_k[\langle \nabla F(\boldsymbol{u}_k), \mathcal{G}_k \rangle] = \mathbb{E}_k \left[ \left\langle \nabla F(\boldsymbol{u}_k), \sum_{i=1}^{N} \boldsymbol{a}_i \boldsymbol{g}_k^{(i)} \right\rangle \right] \tag{92}$$

$$= \left\langle \nabla F(\boldsymbol{u}_k), \sum_{i=1}^{N} \boldsymbol{p}_i \boldsymbol{a}_i \nabla F(\boldsymbol{x}_k^{(i)}) \right\rangle \tag{93}$$

$$= \sum_{i=1}^{N} \boldsymbol{a}_i \left\langle \nabla F(\boldsymbol{u}_k), \boldsymbol{p}_i \nabla F(\boldsymbol{x}_k^{(i)}) \right\rangle \tag{94}$$

$$= \sum_{i=1}^{N} \frac{\boldsymbol{a}_i}{2} \left[ \|\nabla F(\boldsymbol{u}_k)\|^2 + \|\boldsymbol{p}_i \nabla F(\boldsymbol{x}_k^{(i)})\|^2 - \|\nabla F(\boldsymbol{u}_k) - \boldsymbol{p}_i \nabla F(\boldsymbol{x}_k^{(i)})\|^2 \right] \tag{95}$$

$$= \frac{1}{2} \|\nabla F(\boldsymbol{u}_k)\|^2 + \sum_{i=1}^{N} \frac{\boldsymbol{a}_i}{2} \left\| \boldsymbol{p}_i \nabla F(\boldsymbol{x}_k^{(i)}) \right\|^2$$

$$- \sum_{i=1}^{N} \frac{\boldsymbol{a}_i}{2} \left\| \nabla F(\boldsymbol{u}_k) - \boldsymbol{p}_i \nabla F(\boldsymbol{x}_k^{(i)}) \right\|^2 \tag{96}$$

where (93) follows from (20), and (95) follows from the fact that, for arbitrary vectors $\boldsymbol{y}$ and $\boldsymbol{z}$, $2\boldsymbol{y}^T \boldsymbol{z} = \|\boldsymbol{y}\|^2 + \|\boldsymbol{z}\|^2 - \|\boldsymbol{y} - \boldsymbol{z}\|^2$. $\qquad\square$

**Lemma 4.** *Under Assumption 1, following the update rule given in (5), if all model parameters are initialized at the same $\boldsymbol{x}_1$, the expected weighted average gradient is bounded as follows:*

$$\mathbb{E} \left[ \frac{1}{K} \sum_{k=1}^{K} \|\nabla F(\boldsymbol{u}_k)\|^2 \right] \leq \frac{2\left(F(\boldsymbol{x}_1) - F_{inf}\right])}{\eta K} + \sigma^2 \eta L \sum_{i=1}^{N} \boldsymbol{a}_i^2 \boldsymbol{p}_i + \frac{2L^2}{K} \sum_{k=1}^{K} \mathbb{E} \| \boldsymbol{X}_k (\boldsymbol{I} - \boldsymbol{A}) \|_{F_a}^2$$

$$- \frac{1}{K} \sum_{k=1}^{K} \sum_{i=1}^{N} \boldsymbol{a}_i \left( (4\boldsymbol{p}_i - \boldsymbol{p}_i^2 - 2) - \eta L \left( \boldsymbol{a}_i \boldsymbol{p}_i (\beta + 1) - \boldsymbol{a}_i \boldsymbol{p}_i^2 + \boldsymbol{p}_i^2 \right) \right) \mathbb{E} \left\| \nabla F(\boldsymbol{x}_k^{(i)}) \right\|^2 \tag{97}$$

*where $\boldsymbol{A} = \boldsymbol{a} \boldsymbol{1}^T$.*

*Proof.* According to Assumption 1a,

$$\mathbb{E}_k[F(\boldsymbol{u}_{k+1})] - F(\boldsymbol{u}_k) \leq \mathbb{E}_k \left[ \langle \nabla F(\boldsymbol{u}_k) \rangle, \boldsymbol{u}_{k+1} - \boldsymbol{u}_k \rangle + \frac{L}{2} \|\boldsymbol{u}_{k+1} - \boldsymbol{u}_k\|_2^2 \right] \tag{98}$$

$$= -\eta \mathbb{E}_k[\langle \nabla F(\boldsymbol{u}_k), \mathcal{G}_k \rangle] + \frac{\eta^2 L}{2} \mathbb{E}_k \left[ \|\mathcal{G}_k\|_2^2 \right]. \tag{99}$$

Plugging in Lemmas 2 and 3, we get:

$$\mathbb{E}_k[F(\boldsymbol{u}_{k+1})] - F(\boldsymbol{u}_k)$$

$$\leq -\eta \left[ \frac{1}{2} \|\nabla F(\boldsymbol{u}_k)\|^2 + \sum_{i=1}^{N} \frac{\boldsymbol{a}_i}{2} \boldsymbol{p}_i^2 \left\| \nabla F(\boldsymbol{x}_k^{(i)}) \right\|^2 - \sum_{i=1}^{N} \frac{\boldsymbol{a}_i}{2} \|\nabla F(\boldsymbol{u}_k) - \boldsymbol{p}_i \nabla F(\boldsymbol{x}_k^{(i)})\|^2 \right]$$

$$+ \frac{\eta^2 L}{2} \sum_{i=1}^{N} \left( \boldsymbol{a}_i^2 \boldsymbol{p}_i (\beta + 1) - \boldsymbol{a}_i^2 \boldsymbol{p}_i^2 + \boldsymbol{a}_i \boldsymbol{p}_i^2 \right) \left\| \nabla F(\boldsymbol{x}_k^{(i)}) \right\|^2 + \frac{\sigma^2 \eta^2 L}{2} \sum_{i=1}^{N} \boldsymbol{a}_i^2 \boldsymbol{p}_i \tag{100}$$

$$= -\frac{\eta}{2} \|\nabla F(\boldsymbol{u}_k)\|^2 + \frac{\eta}{2} \sum_{i=1}^{N} \boldsymbol{a}_i \|\nabla F(\boldsymbol{u}_k) - \boldsymbol{p}_i \nabla F(\boldsymbol{x}_k^{(i)})\|^2 + \frac{\sigma^2 \eta^2 L}{2} \sum_{i=1}^{N} \boldsymbol{a}_i^2 \boldsymbol{p}_i$$

$$- \frac{\eta}{2} \sum_{i=1}^{N} \boldsymbol{a}_i \left( \boldsymbol{p}_i^2 - \eta L \left( \boldsymbol{a}_i \boldsymbol{p}_i (\beta + 1) - \boldsymbol{a}_i \boldsymbol{p}_i^2 + \boldsymbol{p}_i^2 \right) \right) \left\| \nabla F(\boldsymbol{x}_k^{(i)}) \right\|^2. \tag{101}$$

After some rearranging, we obtain:

$$\|\nabla F(\boldsymbol{u}_k)\|^2 \leq \frac{2\left(F(\boldsymbol{u}_k) - \mathbb{E}_k[F(\boldsymbol{u}_{k+1})]\right)}{\eta} + \sigma^2 \eta L \sum_{i=1}^{N} \boldsymbol{a}_i^2 \boldsymbol{p}_i + \sum_{i=1}^{N} \boldsymbol{a}_i \|\nabla F(\boldsymbol{u}_k) - \boldsymbol{p}_i \nabla F(\boldsymbol{x}_k^{(i)})\|^2$$
$$- \sum_{i=1}^{N} \boldsymbol{a}_i \left(\boldsymbol{p}_i^2 - \eta L \left(\boldsymbol{a}_i \boldsymbol{p}_i (\beta + 1) - \boldsymbol{a}_i \boldsymbol{p}_i^2 + \boldsymbol{p}_i^2\right)\right) \left\|\nabla F(\boldsymbol{x}_k^{(i)})\right\|^2. \tag{102}$$

Taking the total expectation over all iterations:

$$\mathbb{E}\left[\frac{1}{K}\sum_{k=1}^{K}\|\nabla F(\boldsymbol{u}_k)\|^2\right] \leq \frac{2\left(F(\boldsymbol{x}_1) - F_{inf}]\right)}{\eta K} + \sigma^2 \eta L \sum_{i=1}^{N} \boldsymbol{a}_i^2 \boldsymbol{p}_i$$
$$+ \frac{1}{K} \sum_{k=1}^{K} \sum_{i=1}^{N} \boldsymbol{a}_i \mathbb{E}\|\nabla F(\boldsymbol{u}_k) - \boldsymbol{p}_i \nabla F(\boldsymbol{x}_k^{(i)})\|^2$$
$$- \frac{1}{K} \sum_{k=1}^{K} \sum_{i=1}^{N} \boldsymbol{a}_i \left(\boldsymbol{p}_i^2 - \eta L \left(\boldsymbol{a}_i \boldsymbol{p}_i (\beta + 1) - \boldsymbol{a}_i \boldsymbol{p}_i^2 + \boldsymbol{p}_i^2\right)\right) \mathbb{E}\left\|\nabla F(\boldsymbol{x}_k^{(i)})\right\|^2. \tag{103}$$

The third term in (103) can be bounded as:

$$\sum_{i=1}^{N} \boldsymbol{a}_i \mathbb{E}\|\nabla F(\boldsymbol{u}_k) - \boldsymbol{p}_i \nabla F(\boldsymbol{x}_k^{(i)})\|^2$$

$$= \sum_{i=1}^{N} \boldsymbol{a}_i \mathbb{E}\|\nabla F(\boldsymbol{u}_k) - \nabla F(\boldsymbol{x}_k^{(i)}) + (1 - \boldsymbol{p}_i)\nabla F(\boldsymbol{x}_k^{(i)})\|^2 \tag{104}$$

$$\leq \sum_{i=1}^{N} \left[2\boldsymbol{a}_i \mathbb{E}\|\nabla F(\boldsymbol{u}_k) - \nabla F(\boldsymbol{x}_k^{(i)})\|^2 + 2\boldsymbol{a}_i(1 - \boldsymbol{p}_i)^2 \mathbb{E}\|\nabla F(\boldsymbol{x}_k^{(i)})\|^2\right] \tag{105}$$

$$\leq \sum_{i=1}^{N} 2\boldsymbol{a}_i L^2 \mathbb{E}\|\boldsymbol{u}_k - \boldsymbol{x}_k^{(i)}\|^2 + \sum_{i=1}^{N} 2\boldsymbol{a}_i(1 - \boldsymbol{p}_i)^2 \mathbb{E}\|\nabla F(\boldsymbol{x}_k^{(i)})\|^2 \tag{106}$$

where (105) follows from the fact that $\|\boldsymbol{y} + \boldsymbol{z}\|^2 \leq 2\|\boldsymbol{y}\|^2 + 2\|\boldsymbol{z}\|^2$, and (106) follows from (105) by Assumption 1a.

Recalling the definition of the weighted Frobenius norm and the definition of $u$, we can simplify the first term in (106):

$$\sum_{i=1}^{N} 2\boldsymbol{a}_i L^2 \mathbb{E}\|\boldsymbol{u}_k - \boldsymbol{x}_k^{(i)}\|^2 = 2L^2 \mathbb{E}\|\boldsymbol{u}_k \mathbf{1}^T - \mathbf{X}_k\|_{F_{\boldsymbol{a}}}^2 \tag{107}$$

$$= 2L^2 \mathbb{E}\left\|\mathbf{X}_k \boldsymbol{a} \mathbf{1}^T - \mathbf{X}_k\right\|_{F_{\boldsymbol{a}}}^2 \tag{108}$$

$$= 2L^2 \mathbb{E}\|\mathbf{X}_k (\mathbf{I} - \mathbf{A})\|_{F_{\boldsymbol{a}}}^2. \tag{109}$$

Plugging (106) and (109) back into (103), we obtain:

$$
\mathbb{E}\left[\frac{1}{K}\sum_{k=1}^{K}\|\nabla F(\boldsymbol{u}_k)\|^2\right] \leq \frac{2\left(F(\boldsymbol{x}_1) - F_{inf}\right])}{\eta K} + \sigma^2 \eta L \sum_{i=1}^{N} \boldsymbol{a}_i^2 \boldsymbol{p}_i
$$

$$
+ \frac{2L^2}{K}\sum_{k=1}^{K}\mathbb{E}\|\mathbf{X}_k(\mathbf{I}-\mathbf{A})\|_{F_a}^2 + \frac{1}{K}\sum_{k=1}^{K}\sum_{i=1}^{N} 2\boldsymbol{a}_i(1-\boldsymbol{p}_i)^2 \mathbb{E}\|\nabla F(\boldsymbol{x}_k^{(i)})\|^2
$$

$$
- \frac{1}{K}\sum_{k=1}^{K}\sum_{i=1}^{N} \boldsymbol{a}_i \left(\boldsymbol{p}_i^2 - \eta L\left(\boldsymbol{a}_i \boldsymbol{p}_i(\beta+1) - \boldsymbol{a}_i \boldsymbol{p}_i^2 + \boldsymbol{p}_i^2\right)\right) \mathbb{E}\left\|\nabla F(\boldsymbol{x}_k^{(i)})\right\|^2 \tag{110}
$$

$$
= \frac{2\left(F(\boldsymbol{x}_1) - F_{inf}\right])}{\eta K} + \sigma^2 \eta L \sum_{i=1}^{N} \boldsymbol{a}_i^2 \boldsymbol{p}_i + \frac{2L^2}{K}\sum_{k=1}^{K}\mathbb{E}\|\mathbf{X}_k(\mathbf{I}-\mathbf{A})\|_{F_a}^2
$$

$$
- \frac{1}{K}\sum_{k=1}^{K}\sum_{i=1}^{N} \boldsymbol{a}_i \left((4\boldsymbol{p}_i - \boldsymbol{p}_i^2 - 2) - \eta L\left(\boldsymbol{a}_i \boldsymbol{p}_i(\beta+1) - \boldsymbol{a}_i \boldsymbol{p}_i^2 + \boldsymbol{p}_i^2\right)\right) \mathbb{E}\left\|\nabla F(\boldsymbol{x}_k^{(i)})\right\|^2. \tag{111}
$$

$\square$

**Lemma 5.** *Given the properties of $\mathbf{Z}$ and $\mathbf{V}$ given in Propositions 1 and 2, it is the case that:*

$$
\left\|\mathbf{Z}^j - \boldsymbol{A}\right\|_{op} = \zeta^j, \quad \|\boldsymbol{V} - \boldsymbol{A}\|_{op} = 1, \quad \|\boldsymbol{I} - \boldsymbol{A}\|_{op} = 1 \tag{112}
$$

*where $\boldsymbol{A} = \boldsymbol{a}\,\boldsymbol{1}^T$ and $\zeta = \max\{|\lambda_2(\mathbf{H})|, |\lambda(\mathbf{H})|\}$.*

*Proof.* According to the definition of the matrix operator norm,

$$
\left\|\mathbf{Z}^j - \boldsymbol{A}\right\|_{op} = \sqrt{\lambda_{max}((\mathbf{Z}^j - \boldsymbol{A})^T(\mathbf{Z}^j - \boldsymbol{A}))} \tag{113}
$$

$$
= \sqrt{\lambda_{max}(\mathbf{Z}^{2j} - \boldsymbol{A}\,\mathbf{Z}^j - \mathbf{Z}^j\,\boldsymbol{A} + \boldsymbol{A})} \tag{114}
$$

$$
= \sqrt{\lambda_{max}(\mathbf{Z}^{2j} - \boldsymbol{A})} \tag{115}
$$

where (114) follows from $\boldsymbol{A}^j = \boldsymbol{A}$, and (115) follows from $\boldsymbol{A}\,\mathbf{Z} = \mathbf{Z}\,\boldsymbol{A} = \boldsymbol{A}$.

We can simplify (115) further:

$$
= \sqrt{\lambda_{max}(\mathbf{Z}^{2j} - \boldsymbol{A}^{2j})} \tag{116}
$$

$$
= \sqrt{\lambda_{max}(\mathbf{Z} - \boldsymbol{A})^{2j}} \tag{117}
$$

where (117) follows from the commutability of $\mathbf{Z}$ and $\boldsymbol{A}$.

Based on Proposition 2, the non-zero eigenvalues of $\mathbf{Z}$ are the same as $\mathbf{H}$. As shown in Lemma 6 of Rotaru & Nägeli (2004), for a matrix $\mathbf{Z}$ with the properties in Proposition 1, the spectral norm of $\mathbf{Z} - \boldsymbol{A}$ is equal to $\zeta$.

Therefore:

$$
\left\|\mathbf{Z}^j - \boldsymbol{A}\right\|_{op} = \sqrt{\zeta^{2j}} \tag{118}
$$

$$
= \zeta^j. \tag{119}
$$

Similarly for $\mathbf{V}$:

$$
\|\mathbf{V} - \boldsymbol{A}\|_{op} = \sqrt{\lambda_{max}((\mathbf{V} - \boldsymbol{A})^T(\mathbf{V} - \boldsymbol{A}))} \tag{120}
$$

$$
= \sqrt{\lambda_{max}(\mathbf{V} - \boldsymbol{A}\,\mathbf{V} - \mathbf{V}\,\boldsymbol{A} + \boldsymbol{A})} \tag{121}
$$

$$
= \sqrt{\lambda_{max}(\mathbf{V} - \boldsymbol{A})} \tag{122}
$$

$$
\tag{123}
$$

where (121) follows from $\mathbf{A}^j = \mathbf{A}$ and $\mathbf{V}^j = \mathbf{V}$, and (122) follows from $\mathbf{A}\,\mathbf{V} = \mathbf{V}\,\mathbf{A} = \mathbf{A}$.

Note that the eigenvalues of each block $\mathbf{V}^{(d)}$ are $N^{(d)} - 1$ zeros and a one. The set of eigenvalues of $\mathbf{V}$ will include $D$ ones. If $D > 1$, then based on Lemma 6 of Rotaru & Nägeli (2004) and Proposition 1, the spectral norm of $\mathbf{V} - \mathbf{A}$ is 1, so

$$\|\mathbf{V} - \mathbf{A}\|_{op} = \sqrt{1} \tag{124}$$

$$= 1. \tag{125}$$

Since the eigenvalues of $\mathbf{I}$ are all 1, and $\mathbf{I}$ is commutable with $\mathbf{A}$, we can similarly say:

$$\|\mathbf{I} - \mathbf{A}\|_{op} = 1. \tag{126}$$

$\square$

**Lemma 6.** *Given two matrices $\boldsymbol{C} \in \mathbb{R}^{N \times M}$ and $\boldsymbol{D} \in \mathbb{R}^{M \times N}$, and an $N$-vector $\boldsymbol{a}$,*

$$\left| \mathrm{Tr}\!\left( (\mathrm{diag}(\boldsymbol{a}))^{1/2} \, \boldsymbol{C}\boldsymbol{D} \, (\mathrm{diag}(\boldsymbol{a}))^{1/2} \right) \right| \leq \|\boldsymbol{C}\|_{F_a} \|\boldsymbol{D}\|_{F_a}. \tag{127}$$

*Proof.* We define the $i$-th row of $\mathbf{C}$ as $\boldsymbol{c}_i^T$ and the $i$-th column of $\mathbf{D}$ as $\boldsymbol{d}_i$. We can rewrite the trace as:

$$\mathrm{Tr}\!\left( (\mathrm{diag}(\boldsymbol{a}))^{1/2} \, \mathbf{C}\,\mathbf{D} \, (\mathrm{diag}(\boldsymbol{a}))^{1/2} \right) = \sum_{i=1}^{N} \sum_{j=1}^{M} \boldsymbol{a}_i \, \mathbf{C}_{i,j} \, \mathbf{D}_{j,i} \tag{128}$$

$$= \sum_{i=1}^{N} \boldsymbol{a}_i \boldsymbol{c}_i^T \boldsymbol{d}_i. \tag{129}$$

Placing a squared norm around (129), we can apply the Cauchy-Schwartz inequality:

$$\left| \sum_{i=1}^{N} \boldsymbol{a}_i \boldsymbol{c}_i^T \boldsymbol{d}_i \right|^2 \leq \left( \sum_{i=1}^{N} \boldsymbol{a}_i \|\boldsymbol{c}_i^T\|^2 \right) \left( \sum_{i=1}^{N} \boldsymbol{a}_i \|\boldsymbol{d}_i\|^2 \right) \tag{130}$$

$$= \left( \sum_{i=1}^{N} \sum_{j=1}^{M} \boldsymbol{a}_i \, \mathbf{C}_{i,j}^2 \right) \left( \sum_{i=1}^{N} \sum_{j=1}^{M} \boldsymbol{a}_i \, \mathbf{D}_{i,j}^2 \right) \tag{131}$$

$$= \| \mathbf{C} \|_{F_a}^2 \| \mathbf{D} \|_{F_a}^2. \tag{132}$$

$\square$

**Lemma 7.** *Given two matrices $\boldsymbol{C} \in \mathbb{R}^{M \times N}$ and $\boldsymbol{D} \in \mathbb{R}^{N \times N}$, and an $N$-vector $\boldsymbol{a}$, then*

$$\|\boldsymbol{C}\boldsymbol{D}\|_{F_a} \leq \|\boldsymbol{C}\|_{F_a} \|\boldsymbol{D}\|_{op}. \tag{133}$$

*Proof.* We define the $i$-th row of $\mathbf{C}$ as $\boldsymbol{c}_i^T$ and the set $\mathcal{I} = \{i \in [1, M] : \|\boldsymbol{c}_i^T\| \neq 0\}$. We can rewrite the squared Frobenius norm as:

$$\| \mathbf{C}\,\mathbf{D} \|_{F_a}^2 = \sum_{i=1}^{M} \|\boldsymbol{c}_i^T \, \mathbf{D} \, (\mathrm{diag}(\boldsymbol{a}))^{1/2} \|^2 \tag{134}$$

$$= \sum_{i \in \mathcal{I}}^{M} \|\boldsymbol{c}_i^T \, \mathbf{D} \, (\mathrm{diag}(\boldsymbol{a}))^{1/2} \|^2 \tag{135}$$

$$= \sum_{i \in \mathcal{I}}^{M} \|\boldsymbol{c}_i^T \, (\mathrm{diag}(\boldsymbol{a}))^{1/2} \|^2 \frac{\|\boldsymbol{c}_i^T \, \mathbf{D} \, (\mathrm{diag}(\boldsymbol{a}))^{1/2} \|^2}{\|\boldsymbol{c}_i^T \, (\mathrm{diag}(\boldsymbol{a}))^{1/2} \|^2} \tag{136}$$

$$\leq \sum_{i \in \mathcal{I}}^{M} \|\boldsymbol{c}_i^T \, (\mathrm{diag}(\boldsymbol{a}))^{1/2} \|^2 \| \mathbf{D} \|_{op}^2 \tag{137}$$

$$= \| \mathbf{C} \|_{F_a}^2 \| \mathbf{D} \|_{op}^2. \tag{138}$$

$\square$

## C.3 PROOF OF THEOREM 1

We recall Theorem 1.

**Theorem 1.** *Under Assumptions 1 and 2, if $\eta$ satisfies the following for all $i \in \mathcal{M}$:*

$$(4\boldsymbol{p}_i - \boldsymbol{p}_i^2 - 2) \geq \eta L \left( \boldsymbol{a}_i \boldsymbol{p}_i (\beta + 1) - \boldsymbol{a}_i \boldsymbol{p}_i^2 + \boldsymbol{p}_i^2 \right) + 8L^2 \eta^2 q^2 \tau^2 \Gamma \tag{12}$$

*where $\Gamma = \frac{\zeta}{1-\zeta^2} + \frac{2}{1-\zeta} + \frac{\zeta}{(1-\zeta)^2}$ and $\zeta = \max\{|\lambda_2(\mathbf{H})|, |\lambda_N(\mathbf{H})|\}$, then the expected square norm of the average model gradient, averaged over $K$ iterations, is bounded as follows:*

$$\mathbb{E}\left[ \frac{1}{K} \sum_{k=1}^{K} \|\nabla F(\boldsymbol{u}_k)\|_2^2 \right] \leq \frac{2\left( F(\boldsymbol{x}_1) - F_{inf} \right])}{\eta K} + \sigma^2 \eta L \sum_{i=1}^{N} \boldsymbol{a}_i^2 \boldsymbol{p}_i$$

$$+ 4L^2 \eta^2 \sigma^2 q^3 \tau^3 \left( \frac{1}{q\tau} - \frac{1}{K} \right) \left( \frac{\zeta^2}{1-\zeta^2} + \frac{2\zeta}{1-\zeta} + \frac{1}{(1-\zeta)^2} \right) \mathrm{P}$$

$$+ 4L^2 \eta^2 \sigma^2 \left( \frac{2-\zeta}{1-\zeta} \right) \left( \tau^2 \frac{(q-1)(2q+1)}{6} + \frac{(\tau-1)(2\tau+1)}{6} \right) \mathrm{P} \tag{13}$$

$$\xrightarrow{K \to \infty} \quad \sigma^2 \eta L \sum_{i=1}^{N} \boldsymbol{a}_i^2 \boldsymbol{p}_i + 4L^2 \eta^2 \sigma^2 q^2 \tau^2 \left( \frac{\zeta^2}{1-\zeta^2} + \frac{2\zeta}{1-\zeta} + \frac{1}{(1-\zeta)^2} \right) \mathrm{P}$$

$$+ 4L^2 \eta^2 \sigma^2 \left( \frac{2-\zeta}{1-\zeta} \right) \left( \tau^2 \frac{(q-1)(2q+1)}{6} + \frac{(\tau-1)(2\tau+1)}{6} \right) \mathrm{P} \tag{14}$$

*where $\mathrm{P} = \sum_{i=1}^{N} \boldsymbol{a}_i \boldsymbol{p}_i$.*

We now give the proof of Theorem 1 using Lemmas 2-7.

*Proof.* Using our intermediate result from Lemma 4, we decompose $\mathbf{X}_k(\mathbf{I} - \mathbf{A})$ using our recursive definition of $\mathbf{X}_k$:

$$\mathbf{X}_k(\mathbf{I} - \mathbf{A}) = (\mathbf{X}_{k-1} - \eta\, \mathbf{G}_{k-1})\, \mathbf{T}_{k-1}(\mathbf{I} - \mathbf{A}) \tag{139}$$

$$= \mathbf{X}_{k-1}(\mathbf{I} - \mathbf{A})\, \mathbf{T}_{k-1} - \eta\, \mathbf{G}_{k-1}(\mathbf{T}_{k-1} - \mathbf{A}) \tag{140}$$

$$= \left[ (\mathbf{X}_{k-2} - \eta\, \mathbf{G}_{k-2})\, \mathbf{T}_{k-2}(\mathbf{I} - \mathbf{A}) \right] \mathbf{T}_{k-1} - \eta\, \mathbf{G}_{k-1}(\mathbf{T}_{k-1} - \mathbf{A}) \tag{141}$$

$$= \left[ \mathbf{X}_{k-2}(\mathbf{I} - \mathbf{A})\, \mathbf{T}_{k-2} - \eta\, \mathbf{G}_{k-2}(\mathbf{T}_{k-2} - \mathbf{A}) \right] \mathbf{T}_{k-1} - \eta\, \mathbf{G}_{k-1}(\mathbf{T}_{k-1} - \mathbf{A}) \tag{142}$$

$$= \mathbf{X}_{k-2}(\mathbf{I} - \mathbf{A})\, \mathbf{T}_{k-2}\, \mathbf{T}_{k-1} - \eta\, \mathbf{G}_{k-2}(\mathbf{T}_{k-2}\, \mathbf{T}_{k-1} - \mathbf{A}) - \eta\, \mathbf{G}_{k-1}(\mathbf{T}_{k-1} - \mathbf{A}). \tag{143}$$

where (140) follows from the commutability of $\mathbf{T}_k$ and $\mathbf{A}$ by Proposition 4.

Continuing this, we end up with:

$$\mathbf{X}_k(\mathbf{I} - \mathbf{A}) = \mathbf{X}_1(\mathbf{I} - \mathbf{A}) \prod_{l=1}^{k-1} \mathbf{T}_l - \eta \sum_{s=1}^{k-1} \mathbf{G}_s \left( \prod_{l=s}^{k-1} \mathbf{T}_l - \mathbf{A} \right). \tag{144}$$

Since all workers initialize their models to the same vector, $\mathbf{X}_1(\mathbf{I} - \mathbf{A}) \prod_{l=1}^{k-1} \mathbf{T}_k = \mathbf{0}$, and thus we have:

$$\mathbb{E} \left\| \mathbf{X}_k(\mathbf{I} - \mathbf{A}) \right\|_{F_a}^2 = \eta^2 \mathbb{E} \left\| \sum_{s=1}^{k-1} \mathbf{G}_s \left( \prod_{l=s}^{k-1} \mathbf{T}_l - \mathbf{A} \right) \right\|_{F_a}^2. \tag{145}$$

Let $k = jq\tau + l\tau + f$, where $j$ is the number of hub network averaging rounds, $l$ is the number of sub-network averaging rounds since the last hub network averaging round, and $f$ is the number of local iterations since the last sub-network averaging round. Define:

$$\Phi_{s,k-1} = \prod_{l=s}^{k-1} \mathbf{T}_l .$$

Noting that $\mathbf{V}^j = \mathbf{V}$, and $\mathbf{V}\,\mathbf{Z} = \mathbf{Z}\,\mathbf{V} = \mathbf{Z}$ by Proposition 3, $\Phi_{s,k-1}$ can be expressed as:

$$\Phi_{s,k-1} = \begin{cases} \mathbf{I} & jq\tau + l\tau < s < jq\tau + l\tau + f \\ \mathbf{V} & jq\tau < s \leq jq\tau + l\tau \\ \mathbf{Z} & (j-1)q\tau < s \leq jq\tau \\ \mathbf{Z}^2 & (j-2)q\tau < s \leq (j-1)q\tau \\ \vdots \\ \mathbf{Z}^j & 1 \leq s \leq q\tau. \end{cases} \tag{146}$$

For $r < j$, let

$$\mathbf{Y}_r = \sum_{s=rq\tau+1}^{(r+1)q\tau} \mathbf{G}_s\,, \quad \mathbf{Q}_r = \sum_{s=rq\tau+1}^{(r+1)q\tau} \nabla F(\mathbf{X}_s)$$

We also let $\mathbf{Y}_{j_1} = \sum_{s=jq\tau+1}^{jq\tau+l\tau} \mathbf{G}_s$, $\mathbf{Y}_{j_2} = \sum_{s=jq\tau+l\tau+1}^{jq\tau+l\tau+f} \mathbf{G}_s$, $\mathbf{Q}_{j_1} = \sum_{s=jq\tau+1}^{jq\tau+l\tau} \nabla F(\mathbf{X}_s)$, and $\mathbf{Q}_{j_2} = \sum_{s=jq\tau+l\tau+1}^{jq\tau+l\tau+f} \nabla F(\mathbf{X}_s)$. With this in mind, we can split the sum in (145) into batches for each hub network averaging period:

$$\sum_{s=1}^{q\tau} \mathbf{G}_s\left(\Phi_{s,k-1} - \mathbf{A}\right) = \mathbf{Y}_0(\mathbf{Z}^j - \mathbf{A}) \tag{147}$$

$$\sum_{s=q\tau+1}^{2q\tau} \mathbf{G}_s\left(\Phi_{s,k-1} - \mathbf{A}\right) = \mathbf{Y}_1(\mathbf{Z}^{j-1} - \mathbf{A}) \tag{148}$$

$$\ldots$$

$$\sum_{s=(j-1)q\tau+1}^{jq\tau} \mathbf{G}_s\left(\Phi_{s,k-1} - \mathbf{A}\right) = \mathbf{Y}_{j-1}(\mathbf{Z} - \mathbf{A}) \tag{149}$$

$$\sum_{s=jq\tau+1}^{jq\tau+l\tau+f} \mathbf{G}_s\left(\Phi_{s,k-1} - \mathbf{A}\right) = \mathbf{Y}_{j_1}(\mathbf{V} - \mathbf{A}) + \mathbf{Y}_{j_2}(\mathbf{I} - \mathbf{A}). \tag{150}$$

Summing this all together, we get:

$$\sum_{s=1}^{k-1} \mathbf{G}_s\left(\Phi_{s,k-1} - \mathbf{A}\right) = \sum_{r=0}^{j-1} \mathbf{Y}_r(\mathbf{Z}^{j-r} - \mathbf{A}) + \mathbf{Y}_{j_1}(\mathbf{V} - \mathbf{A}) + \mathbf{Y}_{j_2}(\mathbf{I} - \mathbf{A}). \tag{151}$$

Plugging (151) into (145):

$$\mathbb{E}\left\|\mathbf{X}_k(\mathbf{I} - \mathbf{A})\right\|_{F_a}^2 = \eta^2 \mathbb{E}\left\|\sum_{r=0}^{j-1} \mathbf{Y}_r(\mathbf{Z}^{j-r} - \mathbf{A}) + \mathbf{Y}_{j_1}(\mathbf{V} - \mathbf{A}) + \mathbf{Y}_{j_2}(\mathbf{I} - \mathbf{A})\right\|_{F_a}^2 \tag{152}$$

$$= \eta^2 \mathbb{E}\left\|\sum_{r=0}^{j-1}(\mathbf{Y}_r - \mathbf{Q}_r)(\mathbf{Z}^{j-r} - \mathbf{A}) + (\mathbf{Y}_{j_1} - \mathbf{Q}_{j_1})(\mathbf{V} - \mathbf{A})\right.$$

$$\left. + (\mathbf{Y}_{j_2} - \mathbf{Q}_{j_2})(\mathbf{I} - \mathbf{A}) + \sum_{r=0}^{j-1}\mathbf{Q}_r(\mathbf{Z}^{j-r} - \mathbf{A}) + \mathbf{Q}_{j_1}(\mathbf{V} - \mathbf{A}) + \mathbf{Q}_{j_2}(\mathbf{I} - \mathbf{A})\right\|_{F_a}^2 \tag{153}$$

$$\leq 2\eta^2 \underbrace{\mathbb{E}\left\|\sum_{r=0}^{j-1}(\mathbf{Y}_r - \mathbf{Q}_r)(\mathbf{Z}^{j-r} - \mathbf{A}) + (\mathbf{Y}_{j_1} - \mathbf{Q}_{j_1})(\mathbf{V} - \mathbf{A}) + (\mathbf{Y}_{j_2} - \mathbf{Q}_{j_2})(\mathbf{I} - \mathbf{A})\right\|_{F_a}^2}_{T_1}$$

$$+ 2\eta^2 \underbrace{\mathbb{E}\left\|\sum_{r=0}^{j-1}\mathbf{Q}_r(\mathbf{Z}^{j-r} - \mathbf{A}) + \mathbf{Q}_{j_1}(\mathbf{V} - \mathbf{A}) + \mathbf{Q}_{j_2}(\mathbf{I} - \mathbf{A})\right\|_{F_a}^2}_{T_2} \tag{154}$$

where (154) follows from the fact that $\|\boldsymbol{y} + \boldsymbol{z}\|^2 \leq 2\|\boldsymbol{y}\|^2 + 2\|\boldsymbol{z}\|^2$.

We first put a bound on $T_1$:

$$T_1 = 2\eta^2 \mathbb{E} \left\| \sum_{r=0}^{j-1} (\mathbf{Y}_r - \mathbf{Q}_r)(\mathbf{Z}^{j-r} - \mathbf{A}) + (\mathbf{Y}_{j_1} - \mathbf{Q}_{j_1})(\mathbf{V} - \mathbf{A}) + (\mathbf{Y}_{j_2} - \mathbf{Q}_{j_2})(\mathbf{I} - \mathbf{A}) \right\|_{F_{\boldsymbol{a}}}^2 \quad (155)$$

$$= 2\eta^2 \left( \sum_{r=0}^{j-1} \mathbb{E} \left\| (\mathbf{Y}_r - \mathbf{Q}_r)(\mathbf{Z}^{j-r} - \mathbf{A}) \right\|_{F_{\boldsymbol{a}}}^2 + \mathbb{E} \left\| (\mathbf{Y}_{j_1} - \mathbf{Q}_{j_1})(\mathbf{V} - \mathbf{A}) \right\|_{F_{\boldsymbol{a}}}^2 \right.$$

$$\left. + \mathbb{E} \left\| (\mathbf{Y}_{j_2} - \mathbf{Q}_{j_2})(\mathbf{I} - \mathbf{A}) \right\|_{F_{\boldsymbol{a}}}^2 \right)$$

$$+ 2\eta^2 \sum_{n=0}^{j-1} \sum_{l=0,l\neq n}^{j-1} \mathbb{E} \left| \underbrace{\text{Tr}\left( (\text{diag}(\boldsymbol{a}))^{1/2} (\mathbf{Z}^{j-n} - \mathbf{A})(\mathbf{Y}_n - \mathbf{Q}_n)^T (\mathbf{Y}_l - \mathbf{Q}_l)(\mathbf{Z}^{j-l} - \mathbf{A}) (\text{diag}(\boldsymbol{a}))^{1/2} \right)}_{TR} \right|$$
$$\underbrace{\phantom{+ 2\eta^2 \sum_{n=0}^{j-1} \sum_{l=0,l\neq n}^{j-1}}}_{TR_0}$$

$$+ 4\eta^2 \sum_{l=0}^{j-1} \mathbb{E} \underbrace{\left| \text{Tr}\left( (\text{diag}(\boldsymbol{a}))^{1/2} (\mathbf{V} - \mathbf{A})(\mathbf{Y}_{j_1} - \mathbf{Q}_{j_1})^T (\mathbf{Y}_l - \mathbf{Q}_l)(\mathbf{Z}^{j-l} - \mathbf{A}) (\text{diag}(\boldsymbol{a}))^{1/2} \right) \right|}_{TR_1}$$

$$+ 4\eta^2 \sum_{l=0}^{j-1} \mathbb{E} \underbrace{\left| \text{Tr}\left( (\text{diag}(\boldsymbol{a}))^{1/2} (\mathbf{I} - \mathbf{A})(\mathbf{Y}_{j_2} - \mathbf{Q}_{j_2})^T (\mathbf{Y}_l - \mathbf{Q}_l)(\mathbf{Z}^{j-l} - \mathbf{A}) (\text{diag}(\boldsymbol{a}))^{1/2} \right) \right|}_{TR_2}$$

$$+ 4\eta^2 \mathbb{E} \underbrace{\left| \text{Tr}\left( (\text{diag}(\boldsymbol{a}))^{1/2} (\mathbf{V} - \mathbf{A})(\mathbf{Y}_{j_1} - \mathbf{Q}_{j_1})^T (\mathbf{Y}_{j_2} - \mathbf{Q}_{j_2})(\mathbf{I} - \mathbf{A}) (\text{diag}(\boldsymbol{a}))^{1/2} \right) \right|}_{TR_3}. \quad (156)$$

$TR$ can be bounded as:

$$TR \leq \left\| (\mathbf{Z}^{j-n} - \mathbf{A})(\mathbf{Y}_n - \mathbf{Q}_n)^T \right\|_{F_{\boldsymbol{a}}} \left\| (\mathbf{Y}_l - \mathbf{Q}_l)(\mathbf{Z}^{j-l} - \mathbf{A}) \right\|_{F_{\boldsymbol{a}}} \quad (157)$$

$$\leq \left\| (\mathbf{Z}^{j-n} - \mathbf{A}) \right\|_{op} \|\mathbf{Y}_n - \mathbf{Q}_n\|_{F_{\boldsymbol{a}}} \|\mathbf{Y}_l - \mathbf{Q}_l\|_{F_{\boldsymbol{a}}} \left\| (\mathbf{Z}^{j-l} - \mathbf{A}) \right\|_{op} \quad (158)$$

$$\leq \zeta^{2j-n-l} \|\mathbf{Y}_n - \mathbf{Q}_n\|_{F_{\boldsymbol{a}}} \|\mathbf{Y}_l - \mathbf{Q}_l\|_{F_{\boldsymbol{a}}} \quad (159)$$

$$\leq \frac{1}{2} \zeta^{2j-n-l} \left[ \|\mathbf{Y}_n - \mathbf{Q}_n\|_{F_{\boldsymbol{a}}}^2 + \|\mathbf{Y}_l - \mathbf{Q}_l\|_{F_{\boldsymbol{a}}}^2 \right] \quad (160)$$

where (157) follows from Lemma 6, (158) follows from Lemma 7, and (159) follows from Lemma 5. We can similarly bound $TR_1$ and $TR_3$:

$$TR_1 \leq 2\eta^2 \sum_{l=0}^{j-1} \zeta^{j-l} \left[ \mathbb{E} \left\| \mathbf{Y}_{j_1} - \mathbf{Q}_{j_1} \right\|_{F_{\boldsymbol{a}}}^2 + \mathbb{E} \|\mathbf{Y}_l - \mathbf{Q}_l\|_{F_{\boldsymbol{a}}}^2 \right] \quad (161)$$

$$TR_2 \leq 2\eta^2 \sum_{l=0}^{j-1} \zeta^{j-l} \left[ \mathbb{E} \left\| \mathbf{Y}_{j_2} - \mathbf{Q}_{j_2} \right\|_{F_{\boldsymbol{a}}}^2 + \mathbb{E} \|\mathbf{Y}_l - \mathbf{Q}_l\|_{F_{\boldsymbol{a}}}^2 \right] \quad (162)$$

$$TR_3 \leq 2\eta^2 \left[ \mathbb{E} \left\| \mathbf{Y}_{j_1} - \mathbf{Q}_{j_1} \right\|_{F_{\boldsymbol{a}}}^2 + \mathbb{E} \left\| \mathbf{Y}_{j_2} - \mathbf{Q}_{j_2} \right\|_{F_{\boldsymbol{a}}} \right]. \quad (163)$$

Summing $TR_0$ through $TR_3$, we get:

$$\sum_{t=0}^{3} TR_t \leq \eta^2 \sum_{n=0}^{j-1} \sum_{l=0,l\neq n}^{j-1} \zeta^{2j-n-l} \left[ \mathbb{E} \left\| \mathbf{Y}_n - \mathbf{Q}_n \right\|_{F_a}^2 + \mathbb{E} \left\| \mathbf{Y}_l - \mathbf{Q}_l \right\|_{F_a}^2 \right]$$

$$+ 2\eta^2 \sum_{l=0}^{j-1} \zeta^{j-l} \left[ \mathbb{E} \left\| \mathbf{Y}_{j_1} - \mathbf{Q}_{j_1} \right\|_{F_a}^2 + \mathbb{E} \left\| \mathbf{Y}_l - \mathbf{Q}_l \right\|_{F_a}^2 \right]$$

$$+ 2\eta^2 \sum_{l=0}^{j-1} \zeta^{j-l} \left[ \mathbb{E} \left\| \mathbf{Y}_{j_2} - \mathbf{Q}_{j_2} \right\|_{F_a}^2 + \mathbb{E} \left\| \mathbf{Y}_l - \mathbf{Q}_l \right\|_{F_a}^2 \right]$$

$$+ 2\eta^2 \mathbb{E} \left\| \mathbf{Y}_{j_1} - \mathbf{Q}_{j_1} \right\|_{F_a}^2 + 2\eta^2 \mathbb{E} \left\| \mathbf{Y}_{j_2} - \mathbf{Q}_{j_2} \right\|_{F_a}^2 \qquad (164)$$

$$\leq 2\eta^2 \sum_{n=0}^{j-1} \sum_{l=0,l\neq n}^{j-1} \zeta^{2j-n-l} \mathbb{E} \left\| \mathbf{Y}_n - \mathbf{Q}_n \right\|_{F_a}^2$$

$$+ 2\eta^2 \sum_{l=0}^{j-1} \zeta^{j-l} \mathbb{E} \left\| \mathbf{Y}_l - \mathbf{Q}_l \right\|_{F_a}^2 + 2\eta^2 \sum_{l=0}^{j-1} \zeta^{j-l} \mathbb{E} \left\| \mathbf{Y}_l - \mathbf{Q}_l \right\|_{F_a}^2$$

$$+ 2\eta^2 \sum_{l=0}^{j} \zeta^{j-l} \mathbb{E} \left\| \mathbf{Y}_{j_1} - \mathbf{Q}_{j_1} \right\|_{F_a}^2 + 2\eta^2 \sum_{l=0}^{j} \zeta^{j-l} \mathbb{E} \left\| \mathbf{Y}_{j_2} - \mathbf{Q}_{j_2} \right\|_{F_a}^2 \qquad (165)$$

$$= 2\eta^2 \sum_{n=0}^{j-1} \zeta^{j-n} \mathbb{E} \left\| \mathbf{Y}_n - \mathbf{Q}_n \right\|_{F_a}^2 \sum_{l=0,l\neq n}^{j-1} \zeta^{j-l} + 4\eta^2 \sum_{l=0}^{j-1} \zeta^{j-l} \mathbb{E} \left\| \mathbf{Y}_l - \mathbf{Q}_l \right\|_{F_a}^2$$

$$+ 2\eta^2 \mathbb{E} \left\| \mathbf{Y}_{j_1} - \mathbf{Q}_{j_1} \right\|_{F_a}^2 \sum_{l=0}^{j} \zeta^{j-l} + 2\eta^2 \mathbb{E} \left\| \mathbf{Y}_{j_2} - \mathbf{Q}_{j_2} \right\|_{F_a}^2 \sum_{l=0}^{j} \zeta^{j-l} \qquad (166)$$

where (165) follows from the symmetry of the $n$ and $l$ indices.

Plugging (166) back into (156):

$$T_1 \leq 2\eta^2 \sum_{r=0}^{j-1} \mathbb{E} \left\| (\mathbf{Y}_r - \mathbf{Q}_r) \right\|_{F_a}^2 \left\| (\mathbf{Z}^{j-r} - \mathbf{A}) \right\|_{op}^2 + 2\eta^2 \mathbb{E} \left\| (\mathbf{Y}_{j_1} - \mathbf{Q}_{j_1}) \right\|_{F_a}^2 \left\| \mathbf{V} - \mathbf{A} \right\|_{op}^2$$

$$+ 2\eta^2 \mathbb{E} \left\| (\mathbf{Y}_{j_2} - \mathbf{Q}_{j_2}) \right\|_{F_a}^2 \left\| \mathbf{I} - \mathbf{A} \right\|_{op}^2 + 2\eta^2 \sum_{n=0}^{j-1} \zeta^{j-n} \mathbb{E} \left\| \mathbf{Y}_n - \mathbf{Q}_n \right\|_{F_a}^2 \sum_{l=0,l\neq n}^{j-1} \zeta^{j-l}$$

$$+ 2\eta^2 \mathbb{E} \left\| \mathbf{Y}_{j_1} - \mathbf{Q}_{j_1} \right\|_{F_a}^2 \sum_{l=0}^{j} \zeta^{j-l} + 2\eta^2 \mathbb{E} \left\| \mathbf{Y}_{j_2} - \mathbf{Q}_{j_2} \right\|_{F_a}^2 \sum_{l=0}^{j} \zeta^{j-l}$$

$$+ 4\eta^2 \sum_{l=0}^{j-1} \zeta^{j-l} \mathbb{E} \left\| \mathbf{Y}_l - \mathbf{Q}_l \right\|_{F_a}^2 \qquad (167)$$

$$\leq 2\eta^2 \sum_{r=0}^{j-1} \mathbb{E} \left\| (\mathbf{Y}_r - \mathbf{Q}_r) \right\|_{F_a}^2 \zeta^{2(j-r)} + 2\eta^2 \mathbb{E} \left\| (\mathbf{Y}_{j_1} - \mathbf{Q}_{j_1}) \right\|_{F_a}^2$$

$$+ 2\eta^2 \mathbb{E} \left\| (\mathbf{Y}_{j_2} - \mathbf{Q}_{j_2}) \right\|_{F_a}^2 + 2\eta^2 \sum_{n=0}^{j-1} \zeta^{j-n} \mathbb{E} \left\| \mathbf{Y}_n - \mathbf{Q}_n \right\|_{F_a}^2 \sum_{l=0,l\neq n}^{j-1} \zeta^{j-l}$$

$$+ 2\eta^2 \mathbb{E} \left\| \mathbf{Y}_{j_1} - \mathbf{Q}_{j_1} \right\|_{F_a}^2 \sum_{l=0}^{j} \zeta^{j-l} + 2\eta^2 \mathbb{E} \left\| \mathbf{Y}_{j_2} - \mathbf{Q}_{j_2} \right\|_{F_a}^2 \sum_{l=0}^{j} \zeta^{j-l}$$

$$+ 4\eta^2 \sum_{l=0}^{j-1} \zeta^{j-l} \mathbb{E} \left\| \mathbf{Y}_l - \mathbf{Q}_l \right\|_{F_a}^2 \qquad (168)$$

where (167) follows from Lemma 7, and (168) follows from Lemma 5.

We further bound $T_1$:

$$T_1 \leq 2\eta^2 \sum_{r=0}^{j-1} \mathbb{E} \left\| (\mathbf{Y}_r - \mathbf{Q}_r) \right\|_{F_a}^2 \zeta^{2(j-r)} + 2\eta^2 \mathbb{E} \left\| (\mathbf{Y}_{j_1} - \mathbf{Q}_{j_1}) \right\|_{F_a}^2$$

$$+ 2\eta^2 \mathbb{E} \left\| (\mathbf{Y}_{j_2} - \mathbf{Q}_{j_2}) \right\|_{F_a}^2 + 2\eta^2 \sum_{n=0}^{j-1} \zeta^{j-n} \mathbb{E} \left\| \mathbf{Y}_n - \mathbf{Q}_n \right\|_{F_a}^2 \frac{\zeta}{1-\zeta}$$

$$+ 2\eta^2 \mathbb{E} \left\| \mathbf{Y}_{j_1} - \mathbf{Q}_{j_1} \right\|_{F_a}^2 \frac{1}{1-\zeta} + 2\eta^2 \mathbb{E} \left\| \mathbf{Y}_{j_2} - \mathbf{Q}_{j_2} \right\|_{F_a}^2 \frac{1}{1-\zeta}$$

$$+ 4\eta^2 \sum_{l=0}^{j-1} \zeta^{j-l} \mathbb{E} \left\| \mathbf{Y}_l - \mathbf{Q}_l \right\|_{F_a}^2 \tag{169}$$

$$= 2\eta^2 \sum_{r=0}^{j-1} \left( \zeta^{2(j-r)} + 2\zeta^{j-r} + \frac{\zeta^{j-r+1}}{1-\zeta} \right) \mathbb{E} \left\| (\mathbf{Y}_r - \mathbf{Q}_r) \right\|_{F_a}^2$$

$$+ 2\eta^2 \left( \frac{2-\zeta}{1-\zeta} \right) \mathbb{E} \left\| \mathbf{Y}_{j_1} - \mathbf{Q}_{j_1} \right\|_{F_a}^2 + 2\eta^2 \left( \frac{2-\zeta}{1-\zeta} \right) \mathbb{E} \left\| \mathbf{Y}_{j_2} - \mathbf{Q}_{j_2} \right\|_{F_a}^2 \tag{170}$$

where (169) follows from the summation formulae of a power series:

$$\sum_{l=0}^{j} \zeta^{j-l} \leq \sum_{l=-\infty}^{j} \zeta^{j-l} \leq \frac{1}{1-\zeta}, \qquad \sum_{l=0}^{j-1} \zeta^{j-l} \leq \sum_{l=-\infty}^{j-1} \zeta^{j-l} \leq \frac{\zeta}{1-\zeta}. \tag{171}$$

Taking a closer look at $\mathbb{E} \left\| (\mathbf{Y}_r - \mathbf{Q}_r) \right\|_{F_a}^2$ for $0 \leq r < j$:

$$\mathbb{E} \left\| (\mathbf{Y}_r - \mathbf{Q}_r) \right\|_{F_a}^2 = \mathbb{E} \left\| \sum_{s=rq\tau+1}^{(r+1)q\tau} (\mathbf{G}_s - \nabla F(\mathbf{X}_s)) \right\|_{F_a}^2 \tag{172}$$

$$= \sum_{i=1}^{N} \boldsymbol{a}_i \mathbb{E} \left\| \sum_{s=rq\tau+1}^{(r+1)q\tau} (g_s^i - \nabla F(\boldsymbol{x}_s^{(i)})) \right\|^2 \tag{173}$$

$$\leq \sum_{i=1}^{N} \boldsymbol{a}_i q\tau \sum_{s=rq\tau+1}^{(r+1)q\tau} \mathbb{E} \left\| (g_s^i - \nabla F(\boldsymbol{x}_s^{(i)})) \right\|^2 \tag{174}$$

$$\leq q\tau \left( \sum_{i=1}^{N} \boldsymbol{a}_i \sum_{s=rq\tau+1}^{(r+1)q\tau} (\boldsymbol{p}_i(\beta-1)+1) \mathbb{E} \left\| \nabla F(\boldsymbol{x}_s^{(i)}) \right\|^2 \right) + q^2 \tau^2 \sigma^2 \sum_{i=1}^{N} \boldsymbol{a}_i \boldsymbol{p}_i \tag{175}$$

$$= q\tau \left( \sum_{i=1}^{N} \boldsymbol{a}_i \sum_{s=rq\tau+1}^{(r+1)q\tau} (\boldsymbol{p}_i(\beta-1)+1) \mathbb{E} \left\| \nabla F(\boldsymbol{x}_s^{(i)}) \right\|^2 \right) + q^2 \tau^2 \sigma^2 \, \mathrm{P}. \tag{176}$$

where (175) follows from Assumption 1d and (29).

Similarly, for $r = j_1$ and $r = j_2$:

$$\mathbb{E} \left\| (\mathbf{Y}_{j_1} - \mathbf{Q}_{j_1}) \right\|_{F_a}^2 \leq l\tau \left( \sum_{i=1}^{N} \boldsymbol{a}_i \sum_{s=jq\tau+1}^{jq\tau+l\tau} (\boldsymbol{p}_i(\beta-1)+1) \mathbb{E} \left\| \nabla F(\boldsymbol{x}_s^{(i)}) \right\|^2 \right) + l^2 \tau^2 \sigma^2 \, \mathrm{P} \tag{177}$$

$$\mathbb{E} \left\| (\mathbf{Y}_{j_2} - \mathbf{Q}_{j_2}) \right\|_{F_a} \leq (f-1) \left( \sum_{i=1}^{N} \boldsymbol{a}_i \sum_{s=jq\tau+l\tau+1}^{jq\tau+l\tau+f-1} (\boldsymbol{p}_i(\beta-1)+1) \mathbb{E} \left\| \nabla F(\boldsymbol{x}_s^{(i)}) \right\|^2 \right)$$

$$+ (f-1)^2 \sigma^2 \, \mathrm{P}. \tag{178}$$

Plugging (176), (177), and (178) into (170), we can bound $T_1$ as follows:

$$T_1 \leq 2\eta^2\sigma^2\left(\left(q^2\tau^2\sum_{r=0}^{j-1}\left(\zeta^{2(j-r)}+2\zeta^{j-r}+\frac{\zeta^{j-r+1}}{1-\zeta}\right)\right)+\left(\frac{2-\zeta}{1-\zeta}\right)\left(l^2\tau^2+(f-1)^2\right)\right)P$$

$$+2\eta^2q\tau\sum_{r=0}^{j-1}\left(\zeta^{2(j-r)}+2\zeta^{j-r}+\frac{\zeta^{j-r+1}}{1-\zeta}\right)\sum_{s=rq\tau+1}^{(r+1)q\tau}\sum_{i=1}^{N}a_i\left(p_i(\beta-1)+1\right)\mathbb{E}\left\|\nabla F(x_s^{(i)})\right\|^2$$

$$+2\eta^2\left(\frac{2-\zeta}{1-\zeta}\right)l\tau\left(\sum_{i=1}^{N}a_i\sum_{s=jq\tau+1}^{jq\tau+l\tau}\left(p_i(\beta-1)+1\right)\mathbb{E}\left\|\nabla F(x_s^{(i)})\right\|^2\right)$$

$$+2\eta^2\left(\frac{2-\zeta}{1-\zeta}\right)(f-1)\left(\sum_{i=1}^{N}a_i\sum_{s=jq\tau+l\tau+1}^{jq\tau+l\tau+f-1}\left(p_i(\beta-1)+1\right)\mathbb{E}\left\|\nabla F(x_s^{(i)})\right\|^2\right). \tag{179}$$

Referring back to Lemma 4, our goal is to sum $T_1$ over $k=1,\ldots,K$ iterations. First, we sum over the $j$-th sub-network update period up to the $j$-th hub network averaging, for $l=0,\ldots,q-1$ and $f=1,\ldots,\tau$:

$$\sum_{l=0}^{q-1}\sum_{f=1}^{\tau}T_1 \leq 2\eta^2\sigma^2q^3\tau^3\sum_{r=0}^{j-1}\left(\zeta^{2(j-r)}+2\zeta^{j-r}+\frac{\zeta^{j-r+1}}{1-\zeta}\right)P$$

$$+2\eta^2\sigma^2\left(\frac{2-\zeta}{1-\zeta}\right)\left(\tau^3\frac{q(q-1)(2q+1)}{6}+q\frac{\tau(\tau-1)(2\tau+1)}{6}\right)P$$

$$+2\eta^2q^2\tau^2\sum_{r=0}^{j-1}\left(\zeta^{2(j-r)}+2\zeta^{j-r}+\frac{\zeta^{j-r+1}}{1-\zeta}\right)\sum_{s=rq\tau+1}^{(r+1)q\tau}\sum_{i=1}^{N}a_i\left(p_i(\beta-1)+1\right)\mathbb{E}\left\|\nabla F(x_s^{(i)})\right\|^2$$

$$+\eta^2\left(\frac{2-\zeta}{1-\zeta}\right)q(q-1)\tau^2\sum_{i=1}^{N}a_i\sum_{s=jq\tau+1}^{j(q\tau+1)}\left(p_i(\beta-1)+1\right)\mathbb{E}\left\|\nabla F(x_s^{(i)})\right\|^2$$

$$+\eta^2\left(\frac{2-\zeta}{1-\zeta}\right)q^2\tau(\tau-1)\sum_{i=1}^{N}a_i\sum_{s=j(q\tau+1)+1}^{j(q\tau+1)+\tau-1}\left(p_i(\beta-1)+1\right)\mathbb{E}\left\|\nabla F(x_s^{(i)})\right\|^2. \tag{180}$$

Let:

$$\Gamma_r = \left(\zeta^{2(j-r)}+2\zeta^{j-r}+\frac{\zeta^{j-r+1}}{1-\zeta}\right). \tag{181}$$

Note that $\Gamma_j = \frac{3-2\zeta}{1-\zeta} > \frac{2-\zeta}{1-\zeta}$. Using this inequality, we can bound the sum of the last three terms of (180) to get $2q^2\tau^2\sum_{r=0}^{j}\Gamma_r$:

$$\sum_{l=0}^{q-1}\sum_{f=1}^{\tau}T_1 \leq 2\eta^2\sigma^2q^3\tau^3\sum_{r=0}^{j-1}\left(\zeta^{2(j-r)}+2\zeta^{j-r}+\frac{\zeta^{j-r+1}}{1-\zeta}\right)P$$

$$+2\eta^2\sigma^2\left(\frac{2-\zeta}{1-\zeta}\right)\left(\tau^3\frac{q(q-1)(2q+1)}{6}+q\frac{\tau(\tau-1)(2\tau+1)}{6}\right)P$$

$$+2\eta^2q^2\tau^2\sum_{r=0}^{j}\Gamma_r\sum_{s=rq\tau+1}^{(r+1)q\tau}\sum_{i=1}^{N}a_i\left(p_i(\beta-1)+1\right)\mathbb{E}\left\|\nabla F(x_s^{(i)})\right\|^2. \tag{182}$$

Summing (182) over the hub network averaging periods $j = 0, \ldots, K/(q\tau) - 1$, we obtain:

$$\sum_{j=0}^{K/(q\tau)-1} \sum_{l=0}^{q-1} \sum_{f=1}^{\tau} T_1 \leq 2\eta^2\sigma^2 q^3\tau^3 \sum_{j=0}^{K/(q\tau)-1} \sum_{r=0}^{j-1} \left( \zeta^{2(j-r)} + 2\zeta^{j-r} + \frac{\zeta^{j-r+1}}{1-\zeta} \right) \mathrm{P}$$

$$+ 2\eta^2\sigma^2 K \left( \frac{2-\zeta}{1-\zeta} \right) \left( \tau^2 \frac{(q-1)(2q+1)}{6} + \frac{(\tau-1)(2\tau+1)}{6} \right) \mathrm{P}$$

$$+ 2\eta^2 q^2\tau^2 \sum_{j=0}^{K/(q\tau)-1} \sum_{r=0}^{j} \Gamma_r \sum_{s=rq\tau+1}^{(r+1)q\tau} \sum_{i=1}^{N} \boldsymbol{a}_i \left( \boldsymbol{p}_i(\beta-1) + 1 \right) \mathbb{E} \left\| \nabla F(\boldsymbol{x}_s^{(i)}) \right\|^2 \tag{183}$$

$$= 2\eta^2\sigma^2 q^3\tau^3 \sum_{r=0}^{K/(q\tau)-2} \sum_{j=r+1}^{K/(q\tau)-1} \left( \zeta^{2(j-r)} + 2\zeta^{j-r} + \frac{\zeta^{j-r+1}}{1-\zeta} \right) \mathrm{P}$$

$$+ 2\eta^2\sigma^2 K \left( \frac{2-\zeta}{1-\zeta} \right) \left( \tau^2 \frac{(q-1)(2q+1)}{6} + \frac{(\tau-1)(2\tau+1)}{6} \right) \mathrm{P}$$

$$+ 2\eta^2 q^2\tau^2 \sum_{r=0}^{K/(q\tau)-1} \left( \sum_{j=r}^{K/(q\tau)-1} \Gamma_j \right) \left( \sum_{s=rq\tau+1}^{(r+1)q\tau} \sum_{i=1}^{N} \boldsymbol{a}_i \left( \boldsymbol{p}_i(\beta-1) + 1 \right) \mathbb{E} \left\| \nabla F(\boldsymbol{x}_s^{(i)}) \right\|^2 \right).$$
$$\tag{184}$$

Applying the following summation formula to sum over $\Gamma_j$, we obtain

$$\sum_{j=r}^{K/(q\tau)-1} \left( \zeta^{2(j-r)} + 2\zeta^{j-r} + \frac{\zeta^{j-r+1}}{1-\zeta} \right) \leq \sum_{j=r}^{\infty} \left( \zeta^{2(j-r)} + 2\zeta^{j-r} + \frac{\zeta^{j-r+1}}{1-\zeta} \right) \tag{185}$$

$$\leq \frac{1}{1-\zeta^2} + \frac{2}{1-\zeta} + \frac{\zeta}{(1-\zeta)^2}. \tag{186}$$

We let $\Gamma = \frac{1}{1-\zeta^2} + \frac{2}{1-\zeta} + \frac{\zeta}{(1-\zeta)^2}$. We can also apply this following summation formula to the first term in (184):

$$\sum_{j=r+1}^{K/(q\tau)-1} \left( \zeta^{2(j-r)} + 2\zeta^{j-r} + \frac{\zeta^{j-r+1}}{1-\zeta} \right) \leq \sum_{j=r+1}^{\infty} \left( \zeta^{2(j-r)} + 2\zeta^{j-r} + \frac{\zeta^{j-r+1}}{1-\zeta} \right) \tag{187}$$

$$\leq \frac{\zeta^2}{1-\zeta^2} + \frac{2\zeta}{1-\zeta} + \frac{1}{(1-\zeta)^2}. \tag{188}$$

Applying the summation formula in (188), plugging $\Gamma$ in, and indexing the iterations in terms of $k$, we bound (184) as:

$$\sum_{k=1}^{K} T_1 \leq 2\eta^2\sigma^2 q^3\tau^3 \left( \frac{K}{q\tau} - 1 \right) \left( \frac{\zeta^2}{1-\zeta^2} + \frac{2\zeta}{1-\zeta} + \frac{1}{(1-\zeta)^2} \right) \mathrm{P}$$

$$+ 2\eta^2\sigma^2 K \left( \frac{2-\zeta}{1-\zeta} \right) \left( \tau^2 \frac{(q-1)(2q+1)}{6} + \frac{(\tau-1)(2\tau+1)}{6} \right) \mathrm{P}$$

$$+ 2\eta^2 q^2\tau^2 \Gamma \sum_{k=1}^{K} \sum_{i=1}^{N} \boldsymbol{a}_i \left( \boldsymbol{p}_i(\beta-1) + 1 \right) \mathbb{E} \left\| \nabla F(\boldsymbol{x}_k^{(i)}) \right\|^2. \tag{189}$$

Now we bound $T_2$:

$$T_2 = 2\eta^2 \mathbb{E} \left\| \sum_{r=0}^{j-1} \mathbf{Q}_r (\mathbf{Z}^{j-r} - \mathbf{A}) + \mathbf{Q}_{j_1} (\mathbf{V} - \mathbf{A}) + \mathbf{Q}_{j_2} (\mathbf{I} - \mathbf{A}) \right\|_{F_{\boldsymbol{a}}}^2 \tag{190}$$

$$= 2\eta^2 \sum_{r=0}^{j-1} \mathbb{E} \left\| \mathbf{Q}_r (\mathbf{Z}^{j-r} - \mathbf{A}) \right\|_{F_{\boldsymbol{a}}}^2 + 2\eta^2 \mathbb{E} \left\| \mathbf{Q}_{j_1} (\mathbf{V} - \mathbf{A}) \right\|_{F_{\boldsymbol{a}}}^2 + 2\eta^2 \mathbb{E} \left\| \mathbf{Q}_{j_2} (\mathbf{I} - \mathbf{A}) \right\|_{F_{\boldsymbol{a}}}^2$$

$$+ 2\eta^2 \underbrace{\sum_{n=0}^{j-1} \sum_{l=0, l \neq n}^{j-1} \mathbb{E} \left[ \underbrace{\mathrm{Tr} \left( (\mathrm{diag}(\boldsymbol{a}))^{1/2} (\mathbf{Z}^{j-n} - \mathbf{A}) \mathbf{Q}_n^T \mathbf{Q}_l (\mathbf{Z}^{j-l} - \mathbf{A}) (\mathrm{diag}(\boldsymbol{a}))^{1/2} \right)}_{TR'} \right]}_{TR'_0}$$

$$+ 4\eta^2 \underbrace{\sum_{l=0}^{j-1} \mathbb{E} \left[ \mathrm{Tr} \left( (\mathrm{diag}(\boldsymbol{a}))^{1/2} (\mathbf{V} - \mathbf{A}) \mathbf{Q}_{j_1}^T \mathbf{Q}_l (\mathbf{Z}^{j-l} - \mathbf{A}) (\mathrm{diag}(\boldsymbol{a}))^{1/2} \right) \right]}_{TR'_1}$$

$$+ 4\eta^2 \underbrace{\sum_{l=0}^{j-1} \mathbb{E} \left[ \mathrm{Tr} \left( (\mathrm{diag}(\boldsymbol{a}))^{1/2} (\mathbf{I} - \mathbf{A}) \mathbf{Q}_{j_2}^T \mathbf{Q}_l (\mathbf{Z}^{j-l} - \mathbf{A}) (\mathrm{diag}(\boldsymbol{a}))^{1/2} \right) \right]}_{TR'_2}$$

$$+ 4\eta^2 \underbrace{\mathbb{E} \left[ \mathrm{Tr} \left( (\mathrm{diag}(\boldsymbol{a}))^{1/2} (\mathbf{V} - \mathbf{A}) \mathbf{Q}_{j_1}^T \mathbf{Q}_{j_2} (\mathbf{I} - \mathbf{A}) (\mathrm{diag}(\boldsymbol{a}))^{1/2} \right) \right]}_{TR'_3}. \tag{191}$$

$TR'$ can be bounded as:

$$TR' \leq \left\| (\mathbf{Z}^{j-n} - \mathbf{A}) \mathbf{Q}_n^T \right\|_{F_{\boldsymbol{a}}} \left\| \mathbf{Q}_l (\mathbf{Z}^{j-l} - \mathbf{A}) \right\|_{F_{\boldsymbol{a}}} \tag{192}$$

$$\leq \left\| (\mathbf{Z}^{j-n} - \mathbf{A}) \right\|_{op} \left\| \mathbf{Q}_n \right\|_{F_{\boldsymbol{a}}} \left\| \mathbf{Q}_l \right\|_{F_{\boldsymbol{a}}} \left\| (\mathbf{Z}^{j-l} - \mathbf{A}) \right\|_{op} \tag{193}$$

$$\leq \frac{1}{2} \zeta^{2j-n-l} \left[ \left\| \mathbf{Q}_n \right\|_{F_{\boldsymbol{a}}}^2 + \left\| \mathbf{Q}_l \right\|_{F_{\boldsymbol{a}}}^2 \right] \tag{194}$$

where (192) follows from Lemma 6. We can similarly bound $TR'_1$ through $TR'_3$:

$$TR'_1 \leq 2\eta^2 \sum_{l=0}^{j-1} \zeta^{j-l} \left[ \mathbb{E} \left\| \mathbf{Q}_{j_1} \right\|_{F_{\boldsymbol{a}}}^2 + \mathbb{E} \left\| \mathbf{Q}_l \right\|_{F_{\boldsymbol{a}}}^2 \right] \tag{195}$$

$$TR'_2 \leq 2\eta^2 \sum_{l=0}^{j-1} \zeta^{j-l} \left[ \mathbb{E} \left\| \mathbf{Q}_{j_2} \right\|_{F_{\boldsymbol{a}}}^2 + \mathbb{E} \left\| \mathbf{Q}_l \right\|_{F_{\boldsymbol{a}}}^2 \right] \tag{196}$$

$$TR'_3 \leq 2\eta^2 \mathbb{E} \left\| \mathbf{Q}_{j_1} \right\|_{F_{\boldsymbol{a}}}^2 + 2\eta^2 \mathbb{E} \left\| \mathbf{Q}_{j_2} \right\|_{F_{\boldsymbol{a}}}^2. \tag{197}$$

Summing $TR'_0$ through $TR'_3$, we get:

$$\sum_{t=0}^{3} TR'_t \le \eta^2 \sum_{n=0}^{j-1} \sum_{l=0,l\neq n}^{j-1} \zeta^{2j-n-l} \mathbb{E}\left[ \mathbb{E}\|\mathbf{Q}_n\|_{F_a}^2 + \mathbb{E}\|\mathbf{Q}_l\|_{F_a}^2 \right]$$
$$+ 2\eta^2 \sum_{l=0}^{j-1} \zeta^{j-l} \mathbb{E}\|\mathbf{Q}_l\|_{F_a}^2 + 2\eta^2 \sum_{l=0}^{j-1} \zeta^{j-l} \mathbb{E}\|\mathbf{Q}_l\|_{F_a}^2$$
$$+ 2\eta^2 \sum_{l=0}^{j} \zeta^{j-l} \mathbb{E}\|\mathbf{Q}_{j_1}\|_{F_a}^2 + 2\eta^2 \sum_{l=0}^{j} \zeta^{j-l} \mathbb{E}\|\mathbf{Q}_{j_2}\|_{F_a}^2 \quad (198)$$
$$\le 2\eta^2 \sum_{n=0}^{j-1} \zeta^{j-n} \mathbb{E}\|\mathbf{Q}_n\|_{F_a}^2 \sum_{l=0,l\neq n}^{j-1} \zeta^{j-l} + 4\eta^2 \sum_{l=0}^{j-1} \zeta^{j-l} \mathbb{E}\|\mathbf{Q}_l\|_{F_a}^2$$
$$+ 2\eta^2 \mathbb{E}\|\mathbf{Q}_{j_1}\|_{F_a}^2 \sum_{l=0}^{j} \zeta^{j-l} + 2\eta^2 \mathbb{E}\|\mathbf{Q}_{j_2}\|_{F_a}^2 \sum_{l=0}^{j} \zeta^{j-l} \quad (199)$$

where (199) follows from the symmetry of the indices $n$ and $l$.

Plugging (199) back into (191):

$$T_2 \le 2\eta^2 \sum_{r=0}^{j-1} \mathbb{E}\left\|\mathbf{Q}_r(\mathbf{Z}^{j-r}-\mathbf{A})\right\|_{F_a}^2 + 2\eta^2 \mathbb{E}\left\|\mathbf{Q}_{j_1}(\mathbf{V}-\mathbf{A})\right\|_{F_a}^2 + 2\eta^2 \mathbb{E}\left\|\mathbf{Q}_{j_2}(\mathbf{I}-\mathbf{A})\right\|_{F_a}^2$$
$$+ 2\eta^2 \sum_{n=0}^{j-1} \zeta^{j-n} \mathbb{E}\|\mathbf{Q}_n\|_{F_a}^2 \sum_{l=0,l\neq n}^{j-1} \zeta^{j-l} + 4\eta^2 \sum_{l=0}^{j-1} \zeta^{j-l} \mathbb{E}\|\mathbf{Q}_l\|_{F_a}^2$$
$$+ 2\eta^2 \mathbb{E}\|\mathbf{Q}_{j_1}\|_{F_a}^2 \sum_{l=0}^{j} \zeta^{j-l} + 2\eta^2 \mathbb{E}\|\mathbf{Q}_{j_2}\|_{F_a}^2 \sum_{l=0}^{j} \zeta^{j-l} \quad (200)$$
$$\le 2\eta^2 \sum_{r=0}^{j-1} \mathbb{E}\|\mathbf{Q}_r\|_{F_a}^2 \left\|(\mathbf{Z}^{j-r}-\mathbf{A})\right\|_{op}^2 + 2\eta^2 \mathbb{E}\|\mathbf{Q}_{j_1}\|_{F_a}^2 \|\mathbf{V}-\mathbf{A}\|_{op}^2 + 2\eta^2 \mathbb{E}\|\mathbf{Q}_{j_2}\|_{F_a}^2 \|\mathbf{I}-\mathbf{A}\|_{op}^2$$
$$+ 2\eta^2 \sum_{n=0}^{j-1} \zeta^{j-n} \mathbb{E}\|\mathbf{Q}_n\|_{F_a}^2 \sum_{l=0,l\neq n}^{j-1} \zeta^{j-l} + 4\eta^2 \sum_{l=0}^{j-1} \zeta^{j-l} \mathbb{E}\|\mathbf{Q}_l\|_{F_a}^2$$
$$+ 2\eta^2 \mathbb{E}\|\mathbf{Q}_{j_1}\|_{F_a}^2 \sum_{l=0}^{j} \zeta^{j-l} + 2\eta^2 \mathbb{E}\|\mathbf{Q}_{j_2}\|_{F_a}^2 \sum_{l=0}^{j} \zeta^{j-l} \quad (201)$$
$$\le 2\eta^2 \sum_{r=0}^{j-1} \zeta^{j-r} \mathbb{E}\|\mathbf{Q}_r\|_{F_a}^2 + 2\eta^2 \mathbb{E}\|\mathbf{Q}_{j_1}\|_{F_a}^2 + 2\eta^2 \mathbb{E}\|\mathbf{Q}_{j_2}\|_{F_a}^2$$
$$+ 2\eta^2 \sum_{n=0}^{j-1} \zeta^{j-n} \mathbb{E}\|\mathbf{Q}_n\|_{F_a}^2 \sum_{l=0,l\neq n}^{j-1} \zeta^{j-l} + 4\eta^2 \sum_{l=0}^{j-1} \zeta^{j-l} \mathbb{E}\|\mathbf{Q}_l\|_{F_a}^2$$
$$+ 2\eta^2 \mathbb{E}\|\mathbf{Q}_{j_1}\|_{F_a}^2 \sum_{l=0}^{j} \zeta^{j-l} + 2\eta^2 \mathbb{E}\|\mathbf{Q}_{j_2}\|_{F_a}^2 \sum_{l=0}^{j} \zeta^{j-l} \quad (202)$$

where (201) follows from Lemma 7, and (202) follows from Lemma 5.

We further bound $T_2$:

$$
T_2 \leq 2\eta^2 \sum_{r=0}^{j-1} \zeta^{j-r} \mathbb{E} \|\mathbf{Q}_r\|_{F_a}^2 + 2\eta^2 \mathbb{E} \|\mathbf{Q}_{j_1}\|_{F_a}^2 + 2\eta^2 \mathbb{E} \|\mathbf{Q}_{j_2}\|_{F_a}^2
$$

$$
+ 2\eta^2 \sum_{n=0}^{j-1} \zeta^{j-n} \mathbb{E} \|\mathbf{Q}_n\|_{F_a}^2 \frac{\zeta}{1-\zeta} + 4\eta^2 \sum_{l=0}^{j-1} \zeta^{j-l} \mathbb{E} \|\mathbf{Q}_l\|_{F_a}^2
$$

$$
+ 2\eta^2 \mathbb{E} \|\mathbf{Q}_{j_1}\|_{F_a}^2 \frac{1}{1-\zeta} + 2\eta^2 \mathbb{E} \|\mathbf{Q}_{j_2}\|_{F_a}^2 \frac{1}{1-\zeta} \quad (203)
$$

$$
\leq 2\eta^2 \sum_{r=0}^{j-1} \left( \zeta^{2(j-r)} + 2\zeta^{j-r} + \frac{\zeta^{j-r+1}}{1-\zeta} \right) \mathbb{E} \|\mathbf{Q}_r\|_{F_a}^2
$$

$$
+ 2\eta^2 \left( \frac{2-\zeta}{1-\zeta} \right) \mathbb{E} \|\mathbf{Q}_{j_1}\|_{F_a}^2 + 2\eta^2 \left( \frac{2-\zeta}{1-\zeta} \right) \mathbb{E} \|\mathbf{Q}_{j_2}\|_{F_a}^2 \quad (204)
$$

where (203) follows from the summation formulae of a power series in (171).

After applying the definition of $\mathbf{Q}$ to (203), we obtain:

$$
T_2 = 2\eta^2 \sum_{r=0}^{j-1} \left( \zeta^{2(j-r)} + 2\zeta^{j-r} + \frac{\zeta^{j-r+1}}{1-\zeta} \right) \mathbb{E} \left\| \sum_{s=1}^{q\tau} \nabla F(\mathbf{X}_{rq\tau+s}) \right\|_{F_a}^2
$$

$$
+ 2\eta^2 \left( \frac{2-\zeta}{1-\zeta} \right) \mathbb{E} \left\| \sum_{s=1}^{l\tau} \nabla F(\mathbf{X}_{jq\tau+s}) \right\|_{F_a}^2 + 2\eta^2 \left( \frac{2-\zeta}{1-\zeta} \right) \mathbb{E} \left\| \sum_{s=1}^{f-1} \nabla F(\mathbf{X}_{jq\tau+l\tau+s}) \right\|_{F_a}^2 \quad (205)
$$

$$
\leq 2\eta^2 q\tau \sum_{r=0}^{j-1} \left( \zeta^{2(j-r)} + 2\zeta^{j-r} + \frac{\zeta^{j-r+1}}{1-\zeta} \right) \sum_{s=1}^{q\tau} \mathbb{E} \|\nabla F(\mathbf{X}_{rq\tau+s})\|_{F_a}^2
$$

$$
+ 2\eta^2 l\tau \left( \frac{2-\zeta}{1-\zeta} \right) \sum_{s=1}^{l\tau} \mathbb{E} \|\nabla F(\mathbf{X}_{jq\tau+s})\|_{F_a}^2
$$

$$
+ 2\eta^2 (f-1) \left( \frac{2-\zeta}{1-\zeta} \right) \sum_{s=1}^{f-1} \mathbb{E} \|\nabla F(\mathbf{X}_{jq\tau+l\tau+s})\|_{F_a}^2 \quad (206)
$$

where (206) follows from (205) by Jensen's inequality.

Summing over all iterates in the $j$-th sub-network update period, we obtain:

$$
\sum_{l=0}^{q-1} \sum_{f=1}^{\tau} T_2 \leq 2\eta^2 q^2 \tau^2 \sum_{r=0}^{j-1} \left( \left( \zeta^{2(j-r)} + 2\zeta^{j-r} + \frac{\zeta^{j-r+1}}{1-\zeta} \right) \sum_{s=1}^{q\tau} \mathbb{E} \|\nabla F(\mathbf{X}_{rq\tau+s})\|_{F_a}^2 \right)
$$

$$
+ \eta^2 q\tau(q-1) \left( \frac{2-\zeta}{1-\zeta} \right) \sum_{s=1}^{q\tau} \mathbb{E} \|\nabla F(\mathbf{X}_{jq\tau+s})\|_{F_a}^2
$$

$$
+ \eta^2 q\tau(\tau-1) \left( \frac{2-\zeta}{1-\zeta} \right) \sum_{s=1}^{\tau-1} \mathbb{E} \|\nabla F(\mathbf{X}_{jq\tau+q\tau+s})\|_{F_a}^2 \quad (207)
$$

$$
\leq 2\eta^2 q^2 \tau^2 \sum_{r=0}^{j} \Gamma_r \sum_{s=1}^{q\tau} \mathbb{E} \|\nabla F(\mathbf{X}_{rq\tau+s})\|_{F_a}^2 . \quad (208)
$$

Summing over all iterations and applying the summation bound in (186) to (208):

$$
\sum_{j=0}^{K/(q\tau)-1} \sum_{l=0}^{q-1} \sum_{f=1}^{\tau} T_2 \leq 2\eta^2 q^2 \tau^2 \Gamma \sum_{k=1}^{K} \mathbb{E} \|\nabla F(\mathbf{X}_k)\|_{F_a}^2 . \quad (209)
$$

Summing $T_1$ and $T_2$, we obtain

$$\frac{2L^2}{K}\sum_{k=1}^{K}\mathbb{E}\|\mathbf{X}_k(\mathbf{I}-\mathbf{A})\|_{F_a}^2 \le \frac{2L^2}{K}\sum_{k=1}^{K}T_1 + \frac{2L^2}{K}\sum_{k=1}^{K}T_2 \tag{210}$$

$$\le 4L^2\eta^2\sigma^2 q^3\tau^3\left(\frac{1}{q\tau}-\frac{1}{K}\right)\left(\frac{\zeta^2}{1-\zeta^2}+\frac{2\zeta}{1-\zeta}+\frac{1}{(1-\zeta)^2}\right)\mathrm{P}$$

$$+4L^2\eta^2\sigma^2\left(\frac{2-\zeta}{1-\zeta}\right)\left(\tau^2\frac{(q-1)(2q+1)}{6}+\frac{(\tau-1)(2\tau+1)}{6}\right)\mathrm{P}$$

$$+8L^2\eta^2 q^2\tau^2\Gamma\frac{1}{K}\sum_{k=1}^{K}\sum_{i=1}^{N}\boldsymbol{a}_i\left(\boldsymbol{p}_i(\beta-1)+1\right)\mathbb{E}\left\|\nabla F(\boldsymbol{x}_k^{(i)})\right\|^2. \tag{211}$$

Plugging $T_1$ and $T_2$ back into Lemma 4, we arrive at

$$\mathbb{E}\left[\frac{1}{K}\sum_{k=1}^{K}\|\nabla F(\boldsymbol{u}_k)\|^2\right] \le \frac{2\left(F(\boldsymbol{x}_1)-F_{inf}]\right)}{\eta K}+\sigma^2\eta L\sum_{i=1}^{N}\boldsymbol{a}_i^2\boldsymbol{p}_i+\frac{2L^2}{K}\sum_{k=1}^{K}T_1+\frac{2L^2}{K}\sum_{k=1}^{K}T_2$$

$$-\frac{1}{K}\sum_{k=1}^{K}\sum_{i=1}^{N}\boldsymbol{a}_i\left((4\boldsymbol{p}_i-\boldsymbol{p}_i^2-2)-\eta L\left(\boldsymbol{a}_i\boldsymbol{p}_i(\beta+1)-\boldsymbol{a}_i\boldsymbol{p}_i^2+\boldsymbol{p}_i^2\right)\right)\mathbb{E}\left\|\nabla F(\boldsymbol{x}_k^{(i)})\right\|^2 \tag{212}$$

$$=\frac{2\left(F(\boldsymbol{x}_1)-F_{inf}]\right)}{\eta K}+\sigma^2\eta L\sum_{i=1}^{N}\boldsymbol{a}_i^2\boldsymbol{p}_i$$

$$+4L^2\eta^2\sigma^2 q^3\tau^3\left(\frac{1}{q\tau}-\frac{1}{K}\right)\left(\frac{\zeta^2}{1-\zeta^2}+\frac{2\zeta}{1-\zeta}+\frac{1}{(1-\zeta)^2}\right)\mathrm{P}$$

$$+4L^2\eta^2\sigma^2\left(\frac{2-\zeta}{1-\zeta}\right)\left(\tau^2\frac{(q-1)(2q+1)}{6}+\frac{(\tau-1)(2\tau+1)}{6}\right)\mathrm{P}$$

$$-\frac{1}{K}\sum_{k=1}^{K}\sum_{i=1}^{N}\boldsymbol{a}_i\left((4\boldsymbol{p}_i-\boldsymbol{p}_i^2-2)-\eta L\left(\boldsymbol{a}_i\boldsymbol{p}_i(\beta+1)-\boldsymbol{a}_i\boldsymbol{p}_i^2+\boldsymbol{p}_i^2\right)-8L^2\eta^2 q^2\tau^2\Gamma\right)\mathbb{E}\left\|\nabla F(\boldsymbol{x}_k^{(i)})\right\|^2$$

$$\tag{213}$$

If $\eta$ satisfies the following for $i=1,\dots,N$,

$$(4\boldsymbol{p}_i-\boldsymbol{p}_i^2-2)\ge \eta L\left(\boldsymbol{a}_i\boldsymbol{p}_i(\beta+1)-\boldsymbol{a}_i\boldsymbol{p}_i^2+\boldsymbol{p}_i^2\right)+8L^2\eta^2 q^2\tau^2\Gamma \tag{214}$$

then we can simplify (213):

$$\mathbb{E}\left[\frac{1}{K}\sum_{k=1}^{K}\|\nabla F(\boldsymbol{u}_k)\|^2\right] \le \frac{2\left(F(\boldsymbol{x}_1)-F_{inf}]\right)}{\eta K}+\sigma^2\eta L\sum_{i=1}^{N}\boldsymbol{a}_i^2\boldsymbol{p}_i$$

$$+4L^2\eta^2\sigma^2 q^3\tau^3\left(\frac{1}{q\tau}-\frac{1}{K}\right)\left(\frac{\zeta^2}{1-\zeta^2}+\frac{2\zeta}{1-\zeta}+\frac{1}{(1-\zeta)^2}\right)\mathrm{P}$$

$$+4L^2\eta^2\sigma^2\left(\frac{2-\zeta}{1-\zeta}\right)\left(\tau^2\frac{(q-1)(2q+1)}{6}+\frac{(\tau-1)(2\tau+1)}{6}\right)\mathrm{P}. \tag{215}$$

$\square$

## C.4 COMPARISON TO COOPERATIVE SGD

We note that when setting $a_i=1/N$ and $p_i=1$ for all workers $i$, and setting $q=1$, MLL-SGD reduces to Cooperative SGD (Wang & Joshi, 2018). However, the bound in Theorem 1 differs when compared to the bound of Cooperative SGD. Specifically, Theorem 1 has error terms dependent on $\tau^2$ as opposed to $\tau$.

This is due to the formulation of $\boldsymbol{g}_k^{(i)}$. Namely:

$$\mathbb{E}_k[\boldsymbol{g}_k^{(i)}] = \boldsymbol{p}_i\mathbb{E}_k[g(\boldsymbol{x}_k^{(i)})] \tag{216}$$

$$= \boldsymbol{p}_i\nabla F(\boldsymbol{x}_k^{(i)}). \tag{217}$$

Because we cannot assume $p_i = 1$, there are cross terms in the expressions in equations (156) and (173) that do not cancel out. Thus, we needed to use a more conservative analysis at these steps on the proof. This is the reason that plugging in a value of $p_i = 1$ is not enough to recover the same bound as in Cooperative SGD. A similar discrepancy can be observed when comparing with Koloskova et al. (2020).

