# OpenReview forum: "Multi-Level Local SGD: Distributed SGD for Heterogeneous Hierarchical Networks"
_ICLR.cc/2021/Conference — ICLR 2021 Poster_

### Official Review · AnonReviewer2 · 2020-10-27
**Solid theoretical analysis but limited empirical validations**

**Rating:** 7
**Confidence:** 5

**Review:**

## Summary
This paper proposes a new variant of local SGD algorithm to make it be more realistic. In particular, (1) it allows workers to perform different number of local steps, depending on their computational resources; (2) workers are organized in a multi-level structure. Workers connected to one central hub can synchronize frequently and hubs are communicated in an infrequent and decentralized manor.

The authors provide convergence analysis under non-convex, iid data partition settings, and conduct preliminary experiments to validate their theoretical findings.

## Pros
1. This paper is easy to follow and very well-written.
2. It makes a non-trivial extension of (Wang & Joshi, 2018). By introducing probability of taking local updates, the framework allows different workers to take different local steps within a given time interval. This is a realistic setting typically ignored by related literature.
3. The analysis of the proposed algorithm is not trivial. It is nice to see how the authors model the complex algorithm in a simple way, although the formulation is roughly the same as (Wang & Joshi 2018).

## Cons
1. It seems that the results of MTL-SGD cannot recover local SGD? In particular, the additional error terms (the last two terms in (13)) increase with $q^2 \tau^2$. However, in local SGD, the additional error terms increases linearly with $\tau$, as shown in (Wang & Joshi, 2018). I didn't find any discussions on this discrepancy.
2. The experimental results are too limited. Especially, it is hard to see the advantages of MTL-SGD over other two baselines. I encourage the authors to redefine the x-axis in experiments to time slots. Within one time slot, each worker has a probability $p_i$ to perform one local step. In local SGD, in order to finish one round of $\tau$ local updates, the time slots required for one worker is $\tau/p_i$, and hence, the time slots used for one round is $\max_i \tau/p_i$. However, in MTL-SGD, the time slots used per round can be exactly $\tau$ by allowing workers to have different number of local steps. By doing this, MTL-SGD might have much faster convergence than local SGD, in terms of loss versus time slots.

## Post-rebuttal
I've read the authors' response and other reviewers' comments. The response and the updated version clarify my concerns. So I slightly increase my score.

---

> ### Author Response · Authors · 2020-11-22
> **Response to AnonReviewer2**
>
> Thank you again for the thorough review. We have submitted a revision that addresses your comments.
>
> "It seems that the results of MTL-SGD cannot recover local SGD? In particular, the additional error terms (the last two terms in (13)) increase with q2τ2. However, in local SGD, the additional error terms increases linearly with τ, as shown in (Wang & Joshi, 2018). I didn't find any discussions on this discrepancy."
>
> We agree there is a discrepancy, and we have added a paragraph discussing this question at the end of Section 5 on page 7 before Corollary 1 and in Appendix section B.4 on page 35. This discrepancy arises due to the formulation of $g_i$, namely, that $\mathbb{E}[g_i] = p_i \nabla F(x)$.  Because of this, there are cross terms in the expressions in equations in 156 and 173 that do not cancel out. Thus, we needed to use a more conservative analysis at these steps on the proof. This is the reason that plugging in a value of $p_i=1$ is not enough to recover the same bound as in Cooperative SGD.
>
> "The experimental results are too limited. Especially, it is hard to see the advantages of MTL-SGD over other two baselines. I encourage the authors to redefine the x-axis in experiments to time slots. Within one time slot, each worker has a probability pi to perform one local step. In local SGD, in order to finish one round of τ local updates, the time slots required for one worker is τ/pi, and hence, the time slots used for one round is maxi τ/pi. However, in MTL-SGD, the time slots used per round can be exactly τ by allowing workers to have different number of local steps. By doing this, MTL-SGD might have much faster convergence than local SGD, in terms of loss versus time slots."
>
> We thank you for the insightful suggestion as to how to better evaluate MLL-SGD. We have added a new experiment at the end of Section 6 where we compare Local SGD and MLL-SGD with the x-axis measured in time slots as suggested. In every time slot, each worker takes a gradient step with a probability $p_i$. MLL-SGD will wait $\tau$ time slots before averaging, regardless of the number of gradient steps taken, while Local SGD waits for all workers to take $\tau$ gradient steps. This experiment shows that MLL-SGD offers a benefit in terms of convergence time.
> We have also reworked the experiments section as a whole to better illustrate the benefits of the hierarchical model in MLL-SGD with $q>1$, while also exploring the effect of parameters in the algorithm. In addition to the experiments with the CNN, we have included results for CIFAR-10 with ResNet-18 for all experimental settings. The experiments using logistic regression on MNIST have been deferred to Appendix A.

---

### Official Review · AnonReviewer4 · 2020-10-29
**Unsure about how this compares to grouping SGD**

**Rating:** 6
**Confidence:** 4

**Review:**

The paper proposed a multi level SGD algorithm, where workers are assigned to different groups, and each group averages local workers' model.

The idea is a natural extension on existing SGD algorithms (and should be beneficial in heterogeneous networks). The proof should not be hard based on existing asynchronous/decentralized SGD proof. My concern about this paper is that it seems to me the algorithm has a similar motivation with grouping SGD:

W. Jiang et al., "A Novel Stochastic Gradient Descent Algorithm Based on Grouping over Heterogeneous Cluster Systems for Distributed Deep Learning," 2019 19th IEEE/ACM International Symposium on Cluster, Cloud and Grid Computing (CCGRID), Larnaca, Cyprus, 2019, pp. 391-398, doi: 10.1109/CCGRID.2019.00053.

which is a straightforward extension of AllReduce SGD. I suggesting also comparing with it in the experiments. My first thought is that this paper should be faster than grouping-SGD when the network between hubs is not good (and that benefit comes from the fact that communication between hubs in this paper is decentralized).

The hub in this paper does not seem necessary to me also, since it is used to average all local workers' model. Why not just do an AllReduce on all local workers and have one of the worker responsible for averaging with other groups? That should be easier to implement and more efficient.

---

> ### Author Response · Authors · 2020-11-22
> **Response to AnonReviewer4**
>
> Thank you again for the thorough review. We have submitted a revision that addresses your comments.
>
> "My concern about this paper is that it seems to me the algorithm has a similar motivation with grouping SGD:
> W. Jiang et al., "A Novel Stochastic Gradient Descent Algorithm Based on Grouping over Heterogeneous Cluster Systems for Distributed Deep Learning," 2019 19th IEEE/ACM International Symposium on Cluster, Cloud and Grid Computing (CCGRID), Larnaca, Cyprus, 2019, pp. 391-398, doi: 10.1109/CCGRID.2019.00053.
> I suggesting also comparing with it in the experiments. My first thought is that this paper should be faster than grouping-SGD when the network between hubs is not good (and that benefit comes from the fact that communication between hubs in this paper is decentralized)."
>
> We thank you for bringing this paper to our attention. We have added a discussion of Grouping-SGD to the related work section (Section 2 on page 2). After reading about Grouping-SGD in detail, we feel the motivation does differ in a significant way. The goal of the paper by Jiang et al. is to design clusters where workers of similar operating rates or computation power are grouped together. In MLL-SGD, our goal is to design an algorithm that performs well when a communication network structure is imposed a priori. Workers may be clustered due to geographic location, communication constraints, or business policies. The system model also differs as the parameters being trained are vertically (layerwise) partitioned across the parameter servers in Grouping-SGD. Each hub in MLL-SGD, however, is in charge of the full model and updates the entire model in specific training rounds. For these reasons, we feel these algorithms are not comparable. However, we have revised experiments in Section 6 to better illustrate the benefits of MLL-SGD over Local SGD and Hierarchical Local SGD.
>
> "The hub in this paper does not seem necessary to me also, since it is used to average all local workers' model. Why not just do an AllReduce on all local workers and have one of the worker responsible for averaging with other groups? That should be easier to implement and more efficient."
>
> The approach you describe is indeed another way to implement the proposed algorithm. Our analysis is applicable to this approach without modification. We utilize the hub-and-spoke model in each sub-network because this is a standard model in Federated Learning. In typical Federated Learning settings, it is not common for the workers to communicate with each other, hence we have chosen to formulate our problem setting in this specific way.

---

### Official Review · AnonReviewer3 · 2020-10-29
**This paper proposes MLL-SGD for training models in hierarchic networks**

**Rating:** 6
**Confidence:** 3

**Review:**

This paper extends (Wang & Joshi, 2018) and proposes MLL-SGD for training models in hierarchic networks, where the network consists multiple sub-networks, and each sub-network contains multiple workers. In the level of sub-networks, models can be averaged. In the level of workers, the local copies of models can be averaged within a sub-network; however, workers cannot communicate directly with those from a different sub-networks. In such setting, MLL-SGD is proved to enjoy certain convergence property.


# Pros
Hierarchic networks are a common object in practice, but current SGD algorithms fail to cover it due to the hierarchic communication restrictions. Though MLL-SGD itself is a quite naïve extension of existing local SGD methods for handling hierarchic networks, the convergence theory is not a trivial work. I think this paper can serve as a ground work for developing more efficient algorithms for training models in hierarchic networks.


# Cons
- Theorem 1 only considers constant step size and shows a (at least) constant upper bound for MLL-SGD. A theorem that involves decreasing step size and a upper bound that decreasing to zero can be more favorable.

- Eq. (2) and (3). Though here the authors claim they take a probabilistic approach, but actually they treat $tau^(i)$ as constants, where they should be random variables. Can you explain what would be a counterpart of Theorem 1 if $tau^(i)$ are random variables?

- The proof mainly follows (Wang & Joshi, 2018) and the technical contribution is limited. With that being said, I appreciate the formal theory for MLL-SGD in the setting of hierarchic networks.

- Weak experiments. It can be great if there are some experiments that involve training models in real hierarchic networks.


In sum, I think this work is OK but not very significant.

---

> ### Author Response · Authors · 2020-11-22
> **Response to AnonReviewer3**
>
> Thank you again for the thorough review. We have submitted a revision that addresses your comments.
>
> "Theorem 1 only considers constant step size and shows a (at least) constant upper bound for MLL-SGD. A theorem that involves decreasing step size and a upper bound that decreasing to zero can be more favorable."
>
> We agree that analysis of MLL-SGD with a decreasing step size MLL-SGD is an important research question, however, it is not straightforward to extend our analysis to this setting. We have instead added analysis for the case when the convergence rate is constant but is proportional to $1/\sqrt{K}$. This analysis is captured in Corollary 1 at the end of Section 5 on page 7. This result shows that MLL-SGD has the same asymptotic convergence rate as Local SGD and Hierarchical SGD.
>
> "Eq. (2) and (3). Though here the authors claim they take a probabilistic approach, but actually they treat tau(i)  as constants, where they should be random variables. Can you explain what would be a counterpart of Theorem 1 if  tau(i) are random variables?"
>
> We realize the discussion of $\tau_i$ as it relates to $p_i$ may not have been clear. While each $\tau_i$ is constant, the number of local steps a worker takes per global iteration is not, it is a random variable with expected value $tau_i$. We have clarified this definition in the first paragraph of Section 4 on page 4. This model captures a scenario where worker computation rate is relatively stable for the duration of training. We believe that a model where $tau_i$ can change over time is an interesting subject for future work.
>
> "The proof mainly follows (Wang & Joshi, 2018) and the technical contribution is limited. With that being said, I appreciate the formal theory for MLL-SGD in the setting of hierarchic networks."
>
> We thank the reviewer for appreciating the formal theory this paper has on hierarchical networks. We agree the proof follows a similar format to that of (Wang & Joshi, 2018), however we believe that the technical contribution is non-trivial. The addition of heterogeneous worker rates, weighted worker contributions, and multi-level network creates many key differences and difficulties that were not necessary to face in the proof of Cooperative SGD.
>
> "Weak experiments. It can be great if there are some experiments that involve training models in real hierarchic networks."
>
> We understand the importance of testing algorithms in realistic settings. We use synthetic networks so we can evaluate different parameters in isolation. However, we have expanded the experiments in Section 6 to include results with CIFAR-10 and ResNet-18, a more complex dataset and model to address the concern. We also have added a set of experiments to evaluate the convergence time of MLL-SGD against Local SGD which gives an indication of how the algorithms compare in real networks.

---

### Official Review · AnonReviewer1 · 2020-10-29
**initial review**

**Rating:** 6
**Confidence:** 4

**Review:**

The paper extends the idea of hierarchical local SGD by extending the top hierarchy level to decentralized communication. The multi-level local SGD is achieved by the edge devices (performing all-reduce per \tau local edge steps in terms of the corresponding hub) and hubs (gossip averaging per q local hub steps with the neighboring hubs).

### pros
* the paper considers a probabilistic form of the local edge update, making the whole system formulation more realistic.
* A convergence analysis is given for the provided system formulation.

### cons
1. the formulated problem can be seen as a decentralized optimization problem with specifically designed time-varying communication topology and local edge update steps. It is better to compare the derived rate with the existing rate (e.g. in [1]) for the case of $p_i=1$, otherwise, it is unclear the tightness of the derived rate.
2. the claim 'if the gradient of F has a large Lipschitz constant, the step size in the algorithm is large' looks strange to me; normally we have stepsize <= 1/L.
3. the motivation example of the system is formed by edges devices and data-center hubs. however, in both theoretical analysis and numerical results, only the iid data case is been investigated. It somehow contrasts motivation.
4. the numerical evaluation setup might be unfair.
    1. the evaluated decentralized communication over the hubs is unclear to me and why it is a manually generated graph with $\zeta=0.74$.
    2. only the learning curves w.r.t. update step k are visualized. due to the extra gossip averaging steps over hubs, the current observations (i.e. improved convergence speed over local SGD) are reasonable but it is hard to identify the exact benefits of the proposed algorithm. is it possible to use a communication model to simulate some important metrics, e.g. time-to-target-loss? is it possible to include the numerical results of hierarchical local SGD e.g. in Lin et al (i.e. All-reduce is performed for hubs) as an extra baseline?
5. The provided numerical results are limited to the toy problems and may not be sufficient to justify the effectiveness of the proposed algorithm, due to the existing large amount of local SGD work (either empirical or theoretical).

### reference
1. A Unified Theory of Decentralized SGD with Changing Topology and Local Updates, ICML 2020.

---

> ### Author Response · Authors · 2020-11-14
> **Clarification question**
>
> Thank you for your thorough review. We plan on addressing all your concerns in a revision of the paper. We had a question of clarification pertaining to your final point. By “toy problems” are you referring to the model and the dataset used?

---

> > ### Comment · AnonReviewer1 · 2020-11-14
> > **clarify "toy problems"**
> >
> > The paper only evaluated the logistic regression model on MNIST and (a simple) CNN model on EMNIST; these empirical results are not sufficient to justify the trade-off (communication v.s. training/test performance) of the proposed multi-level local SGD.
> >
> > It is suggested to include some results for training standard neural networks (e.g. ResNet) on relative challenge datasets (e.g. CIFAR).

---

> ### Author Response · Authors · 2020-11-22
> **Response to AnonReviewer1 (1/2)**
>
> Thank you again for the thorough review. We have submitted a revision that addresses your comments.
>
> "The formulated problem can be seen as a decentralized optimization problem with specifically designed time-varying communication topology and local edge update steps. It is better to compare the derived rate with the existing rate (e.g. in [1]) for the case of pi=1, otherwise, it is unclear the tightness of the derived rate."
>
> In thoroughly examining the bound in [1], we agree that there is discrepancy with our bound. Specifically, there is an extra factor of $\tau$. This discrepancy arises due to the formulation of $g_i$, namely, that $\mathbb{E}[g_i] = p_i \nabla F(x)$.  Because of this, there are cross terms in the expressions in equations in 156 and 173 that do not cancel out. Thus, we needed to use a more conservative analysis at these steps on the proof. This is the reason that plugging in a value of $p_i=1$ is not enough to recover the same bound as in [1]. We note that a reference [2] shows a similar discrepancy to the bound in [1] as our own results.
> We have added a paragraph discussing this question at the end of Section 5 on page 7 before Corollary 1 and in Appendix section B.4 on page 35.
>
> References
>
> [1] A Unified Theory of Decentralized SGD with Changing Topology and Local Updates, ICML 2020.
>
> [2] J. Wang and G. Joshi. Cooperative sgd: A unified framework for the design and analysis of
> communication-efficient sgd algorithms. arXiv preprint arXiv:1808.07576, 2018.
>
> "the claim 'if the gradient of F has a large Lipschitz constant, the step size in the algorithm is large' looks strange to me; normally we have stepsize <= 1/L."
>
> We understand that the construction of this sentence is confusing since $\eta$ and $L$ are typically inversely proportional, and we have removed this sentence.
>
> "the motivation example of the system is formed by edges devices and data-center hubs. however, in both theoretical analysis and numerical results, only the iid data case is been investigated. It somehow contrasts motivation."
>
> We agree that this assumption that the data distribution is IID may not apply to all scenarios. However, we believe our work lays an important foundation for convergence analysis in multi-level communication networks and where workers have heterogeneous operating rates. We believe that extending this analysis to data that is non-IID is an important open question as we state in the conclusion.

---

> > ### Author Response · Authors · 2020-11-22
> > **Response to AnonReviewer1 (2/2)**
> >
> > "the numerical evaluation setup might be unfair.
> > the evaluated decentralized communication over the hubs is unclear to me and why it is a manually generated graph with ζ=0.74"
> >
> > We realize the choice of this hub network graph was not well-explained. A complete graph will have $\zeta=0$, and a path graph (worst-case connected graph) has a $\zeta$ on the order of $1-\frac{1}{N}$. We selected that hub graph with zeta = 0.74 as a happy medium. However, we have revised our experiments and no longer use this graph in the section. We use a complete graph between hubs when comparing MLL-SGD with Local SGD, since MLL-SGD and Local SGD are equivalent when there is a complete hub graph and $q=1$. In the first experiment shown in Figure 1, we increase the value of $q$ in MLL-SGD while keeping $q\tau$ the same to show its benefit over Local SGD.
> >
> > "only the learning curves w.r.t. update step k are visualized. due to the extra gossip averaging steps over hubs, the current observations (i.e. improved convergence speed over local SGD) are reasonable but it is hard to identify the exact benefits of the proposed algorithm. is it possible to use a communication model to simulate some important metrics, e.g. time-to-target-loss? is it possible to include the numerical results of hierarchical local SGD e.g. in Lin et al (i.e. All-reduce is performed for hubs) as an extra baseline?"
> >
> > We appreciate the suggestion to explore the convergence speed in the experiments. We have added a new experiment at the end of Section 6 in Figure 4 where we compare Hierarchical Local SGD and MLL-SGD with the x-axis measured in “time slots.” In every time slot, each worker will take a gradient step with a probability $p_i$. MLL-SGD will wait $\tau$ time slots before averaging, regardless of the number of gradient steps taken, while Hierarchical Local SGD waits for all workers to take $\tau$ gradient steps. This allows us to compare the progress of each algorithm over time. Our results show a benefit to MLL-SGD over Hierarchical Local SGD in terms of convergence time.
> >
> > "The provided numerical results are limited to the toy problems and may not be sufficient to justify the effectiveness of the proposed algorithm, due to the existing large amount of local SGD work (either empirical or theoretical)."
> >
> > We understand your concern and have completely reworked the experiments section (Section 6). We now include results using a CNN with EMNIST, as well as ResNet-18 with CIFAR-10 for all of the experimental settings. Experiments using logistic regression on MNIST have been deferred to the Appendix A. We have also added experiments to explicitly study the convergence time, as described above.

---

> ### Comment · AnonReviewer1 · 2020-11-24
> **post-rebuttal**
>
> Thank you for the authors' responses and significant revision. The responses have addressed most of my concerns thus I will increase my score from 5 to 6. However, the authors are encouraged to further address the following points:
> 1. what is the difference between "$\tau$ time slots" and "$\tau$ gradient steps", regarding Figure 6? It would be great if the authors can include more simulation details, or release the source code.
> 2. Is it possible to study the impact of different $\tau q$ values for the proposed MLL-SGD?

---

> > ### Author Response · Authors · 2020-11-24
> > **re: post-rebuttal**
> >
> > Thank you for taking your time to review our revision. We have submitted another revision to address your new comments.
> >
> > “What is the difference between "τ time slots" and "τ gradient steps", regarding Figure 6? It would be great if the authors can include more simulation details, or release the source code.”
> >
> > We have added a few sentences to the second to last paragraph of Section 6 on page 9 to better explain the difference between time slots and gradient steps. In every time slot, each worker will take a gradient step with a probability $p_i$. When $p_i = 1$ for a worker $i$, the number of gradient steps taken will match the number of time slots $T$. Otherwise, the number of gradient steps taken will be $T \cdot p_i$ in expectation.
> >
> > We will work towards preparing our code for release on Github.
> >
> > “Is it possible to study the impact of different τq values for the proposed MLL-SGD?”
> >
> > Due to time constraints, we were unable to explore more values of $\tau q$, though we agree it would be interesting to explore in future work
> >
> > We also note that when reviewing the experiments we noticed a minor plotting error. The changes to the plots are negligible, but for completeness we have updated Figure 6 in the new submission.

---

### Decision · Program_Chairs · 2021-01-07
**Final Decision**

**Decision:**

Accept (Poster)

**Comment:**

The paper studies a hierarchical or multi-level version of local SGD, extending earlier work by (Wang & Joshi, 2018), (Lin et al, 2018) and  (Jiang et al. 2019) among others. It gives novel convergence rates in relevant settings, such as by allowing different workers to take different numbers of local steps within a given time interval. The current analysis is restricted to the IID data case, but still insightful, and might serve as a useful building block for follow-up research in the future.
Smaller concerns remained that the presented multi-level results cannot exactly recover local SGD as a special case. Nevertheless the consensus remained that the overall contributions and relevance of the paper remain above the bar. In the discussion phase, several concerns were clarified and additional deep learning experiments have been added to the paper, which is appreciated.